# DAG-Based Column Generation for Adversarial Team Games

## Abstract

Many works recently have focused on computing optimal solutions for the ex ante coordination of a team for solving sequential adversarial team games, where a team of players coordinate against an opponent (or a team of players) in a zero-sum extensive-form game. However, it is challenging to directly compute such an optimal solution because the team's coordinated strategy space is exponential in the size of the game tree due to the asymmetric information of team members. Column Generation (CG) algorithms have been proposed to overcome this challenge by iteratively expanding the team's coordinated strategy space via a Best Response Oracle (BRO). More recently, more compact representations (particularly, the Team Belief Directed Acyclic Graph (TB-DAG)) of the team's coordinated strategy space have been proposed, but the TB-DAG-based algorithms only outperform the CG-based algorithms in games with a small TB-DAG. Unfortunately, it is inefficient to directly apply CG to the TB-DAG because the size of the TB-DAG is still exponential in the size of the game tree and then makes the BRO unscalable. To this end, we develop our novel TB-DAG CG (DCG) algorithm framework by computing a coordinated best response in the original game first and then transforming this strategy into the TB-DAG form. To further improve the scalability, we propose a more suitable BRO for DCG to reduce the cost of the transformation at each iteration. We theoretically show that our algorithm converges exponentially faster than the state-of-the-art CG algorithms, and experimental results show that our algorithm is at least two orders of magnitude faster than the state-of-the-art baselines and solves games that were previously unsolvable.

## 1 Introduction

Many research efforts on computational game theory have focused on computing a Nash equilibrium (Nash, 1951) in two-player zero-sum Extensive-Form Games (EFGs) (Zinkevich et al., 2008; Moravčík et al., 2017; Brown & Sandholm, 2018; Zhang & Sandholm, 2020), where two players receive opposite payoffs. In this setting, a Nash equilibrium can be computed in polynomial time in the size of the EFG (Shoham & Leyton-Brown, 2008). Recent landmark results such as superhuman performance in the heads-up no-limit Texas hold'em poker game (Moravčík et al., 2017; Brown & Sandholm, 2018) show that researchers have understood the problem of computing a Nash equilibrium in two-player zero-sum EFGs well in both theory and practice. However, such a problem in multiplayer games is not well understood, where computing a Nash equilibrium is generally hard (Chen & Deng, 2005). Despite the progress made in multiagent systems due to the development of large language models (LLMs) (Park et al., 2023; Li et al., 2023; Xi et al., 2023), these LLMs or existing multiagent learning algorithms (Lowe et al., 2017; Lanctot et al., 2017; Rashid et al., 2018; Yu et al., 2022; Zhong et al., 2023) that can be used with LLMs cannot guarantee to achieve an optimal solution in multiagent (more than two agents) systems, even if they can guarantee to achieve a stationary point (i.e., a local Nash equilibrium).

In this paper, we focus on sequential Adversarial Team Games (ATG), where a team of players coordinate against an opponent (or a team of players) in an EFG. Specifically, we focus on the solution concept called Team-Maxmin Equilibrium with Coordination device (TMECor) (Celli & Gatti, 2018), which models the ex ante coordination of team members who share the same payoff function. That is, the team members agree on a common strategy before the game starts, but they cannot communicate during playing the game in face of private information for each team member.

Examples of this setting include collusion in poker games and a team of drones playing against an intruder (Carminati et al., 2022). Even though a TMECor is equivalent to a Nash equilibrium in a two-player (a team and an opponent) zero-sum game, computing a TMECor is hard (i.e., APX-hard) (Celli & Gatti, 2018). The main barrier is the team members' asymmetric information in this setting, which makes a team of players equivalent to a single player with imperfect recall and then the behavioral strategy space of the team is not realization equivalent to the normal-form strategy space of the team (Kuhn, 1953). The normal-formal strategies of the team could be arbitrarily better than the behavioral strategies of the team, but the normal-formal strategy space is exponential in the size of the game tree in an ATG.

To efficiently compute a TMECor, some approaches have been proposed. Even though computing a TMECor can be formulated as a linear program (Celli & Gatti, 2018), its size is exponential in the size of the game tree due to the exponential explosion of the team's joint normal-formal strategy (i.e., **coordinated strategy**) space. Column Generation (CG) algorithms (McMahan et al., 2003; Zhang & An, 2020; Farina et al., 2021) were proposed to overcome this challenge (Celli & Gatti, 2018; Zhang et al., 2021; Farina et al., 2021; Zhang et al., 2022b) by iteratively expanding the team's coordinated strategy space via a Best Response Oracle (BRO) (i.e., **normal-form CG**). More recently, a generalization of the sequence form for the team via the tree decomposition was proposed (Zhang & Sandholm, 2022) as a compact representation for the team's coordinated strategy space, and another similar representation (Carminati et al., 2022) was proposed to capture the public information of the team. The Team Belief Directed Acyclic Graph (TB-DAG) representation (Zhang et al., 2022b;c) was proposed to unify the previous two representations, which is a decision problem of the team in an ATG. However, the TB-DAG-based algorithms only outperform the CG-based algorithms in games with a small TB-DAG. To solve games more efficiently, one straightforward idea is applying CG to the TB-DAG. That is, we compute a TMECor with a limited size of the TB-DAG for the restricted game and then expand the TB-DAG by computing a best response over the whole TB-DAG of the original game. Unfortunately, it is inefficient to compute a best response over the whole exponential-sized TB-DAG. Therefore, the CG directly applied to the TB-DAG is inefficient.

To this end, we propose our novel TB-DAG CG (**DCG**) algorithm framework. DCG first computes a coordinated best response in the original game tree, which is represented by a joint normal-form strategy of the team. This best response is then transformed into the TB-DAG form. DCG is inspired by the following two observations:

1. By exploiting the team's correlation property to solve the BRO's integer program faster, the state-of-the-art BRO (Zhang et al., 2021; Farina et al., 2021) can be used in our DCG to significantly outperform the BRO computing a best response over the whole exponential-sized TB-DAG without such a correlation property. As a result, the DCG should significantly outperform the CG directly applied to the TB-DAG.

2. Intuitively, this DCG should not be more efficient than the normal-form CG, as it requires an extra step for transformation at each iteration. However, we show that the TB-DAG formed by a set of TB-DAG form strategies, which are transformed from a set of coordinated strategies, can represent new coordinated strategies due to the new combinations of states and actions in this TB-DAG. This property makes DCG converge in significantly fewer iterations than the normal-form CG in large games. Then DCG outperforms the normal-form CG when the benefit from reducing the number of interactions for convergence surpasses the cost of the transformation.

Unfortunately, DCG suffers from a very high cost of transformation in large games if the coordinated best response is computed by the prior state-of-the-art BRO (Zhang et al., 2021; Farina et al., 2021), as it involves randomized strategies and thus induces a large TB-DAG. To further improve the scalability, we propose a more suitable BRO for DCG to reduce the cost of the transformation. That is, we propose an efficient pure BRO to compute a coordinated best response with a pure strategy for each team member, which will ensure that the corresponding TB-DAG form is small enough.

We theoretically show that our DCG converges exponentially faster than the normal-form CG shown in Zhang et al. (2021). Moreover, experimental results show that our DCG is at least two orders of magnitude faster than the state-of-the-art baselines and solves games that were previously unsolvable. Thus, this paper provides the first efficient TB-DAG CG algorithm. In addition, this paper creates

a fundamental theory for applying the multiagent learning framework – Policy-Spaced Response Oracles (PSRO) (Lanctot et al., 2017) (a variant of CG) – to the TB-DAG for a TMECor.

## 2 PRELIMINARIES

**Adversarial Team Games.** The Extensive-Form Game (EFG) with imperfect information (Shoham & Leyton-Brown, 2008) models the interactions among players through game trees (e.g., Figure 1(a)). Given a set of players $N$ and the chance player $c$ used to model the stochastic events (e.g., drawing cards in poker), an EFG defines a tree through a tuple $\langle N \cup \{c\}, H, Z, A, \{u_i\}_{i \in N} \rangle$, where $H$ is the set of nonterminal nodes of players in $N \cup \{c\}$, and $Z$ is the set of leaf nodes (i.e., terminal nodes). This paper focuses on Adversarial Team Games (ATGs), where $N = T \cup \{o\}$ with that a team of players $T$ with $|T| \geq 2$ plays against an opponent (or a team of players) $o$. $A = \cup_{i \in N \cup \{c\}} A_i$ is the set of all the possible edges (i.e., actions) in the tree, where $A_i$ is the set of player $i$'s actions. $A(h)$ is the set of actions available at node $h \in H$, and $H_i$ is the set of nodes with the acting player $i$ with $H_T = \cup_{i \in T} H_i$. $u_i : Z \to \mathbb{R}$ is player $i$'s utility function that assigns a utility to each leaf node. In an ATG, $u_i = u_j$ for all $i$ and $j$ in $T$, and we denote $u_T = \sum_{i \in T} u_i$. We focus on zero-sum ATGs, where $u_T = -u_o$.

Information sets that are partitions of nonterminal nodes are used to model imperfect (private) information. An information set $I_i$ (i.e., private state) of player $i$ is a set of nodes that are indistinguishable to player $i$, and $\mathcal{I}_i$ is the set of player $i$'s information sets. $\mathcal{I}_T = \cup_{i \in T} \mathcal{I}_i$ is the set of information sets of team $T$. $I_i$'s action set is $A(I_i)$, and $A(I_i) = A(h)$ for each $h \in I_i$, i.e., the nodes in $I_i$ have the same set of actions. For two nodes $h$ and $h'$, $h \preceq h'$ represents that there is a path in the game tree from $h$ to $h'$, and $h \prec h'$ if $h \neq h'$ and $h \preceq h'$. Similarly, $h \preceq I_i$ if there is a node $h'$ in $I_i$ such that $h \preceq h'$. Two information sets $I_i$ and $I_j$ are connected, denoted by $I_i \rightleftharpoons I_j$, if there are $h \in I_i$ and $h' \in I_j$ such that $h \preceq h'$ or $h' \preceq h$. For example, in Figure 1(a), $b \preceq d$, and $b \preceq I_2$ (node $b$ also represents an information set). We focus on games with perfect recall for each player, where players do not forget information. The team as a whole may have imperfect recall due to potentially asymmetric information among team members. After treating the team as a whole, the EFG in this paper can be formulated as a two-player imperfect-recall game.

**Sequences.** For each node $h \in H \cup Z$, the ordered set of player $i$'s actions on the path from the root to $h$ can be defined by a sequence $\sigma_i$. $\Sigma_i = \{(I_i, a_i) : I_i \in \mathcal{I}_i, a_i \in A(I_i)\} \cup \{\varnothing\}$ is the set of player $i$'s sequences, where $\varnothing$ is the empty sequence. $\sigma_i(h)$ is $h$'s parent sequence, which is the last sequence of player $i$ on the path from the root to $h$. For each $I_i \in \mathcal{I}_i$, $I_i$'s parent sequence is $\sigma_i(I_i)$, and $\sigma_i(I_i) = \sigma_i(h)$ for each $h \in I_i$ due to the perfect recall of player $i$. In addition, let $\sigma_i(I_i) = \varnothing$ if there is no information set of player $i$'s before $I_i$. $\boldsymbol{\Sigma}_{N'} = \times_{i \in N' \subseteq N} \Sigma_i$ defines the combinations of sequences $\boldsymbol{\sigma}_{N'}$ of players in $N' \subseteq N$, $\boldsymbol{\sigma}_{N'}[j]$ is the sequence of player $j$ in the joint sequence $\boldsymbol{\sigma}_{N'} \in \boldsymbol{\Sigma}_{N'}$. $\boldsymbol{\sigma}_{N'}(h)$ is the joint sequence of $N'$ reaching $h$. Two sequences $\sigma_i \in \Sigma_i$ and $\sigma_j \in \Sigma_j$ are relevant if any of them is $\varnothing$, or $I_i \rightleftharpoons I_j$ with $\sigma_i = (I_i, a_i)$ and $\sigma_j = (I_j, a_j)$, denoted by $\sigma_i \bowtie \sigma_j$. A joint sequence $\boldsymbol{\sigma}_{N'}$ is relevant if, for any $i, j \in N'$, $\boldsymbol{\sigma}_{N'}[i]$ and $\boldsymbol{\sigma}_{N'}[j]$ in $\boldsymbol{\sigma}_{N'}$ are relevant. $\textcolor{red}{\boldsymbol{\Sigma}_{N'}^{\bowtie} = \{\boldsymbol{\sigma}_{N'}(h_i) \mid h_i \in H \cup Z\} \cup \{(\sigma_i, \times_{j \in N' \setminus \{i\}} \varnothing) \mid \sigma_i \in \Sigma_i, i \in N'\}}$ defines a set of relevant joint sequences. $\textcolor{red}{\text{For } i \notin N' \subseteq N, I_i \bowtie \boldsymbol{\sigma}_{N'} \text{ defines that } \boldsymbol{\sigma}_{N'} \text{ is relevant to } I_i \text{ if } (\sigma_i(I_i), \boldsymbol{\sigma}_{N'}) \in \boldsymbol{\Sigma}_{\{i\} \cup N'}^{\bowtie}.}$

**Reduced-normal-form plans.** A reduced-normal-form plan $\pi_i$ of player $i$ defines an action for every reachable information set $I_i \in \mathcal{I}_i$ due to earlier actions. We use $\pi_i(I_i, a_i) = 1$ to represent that $I_i$ is reachable and $a_i \in A(I_i)$ is played in $\pi_i$. Specifically, $\pi_i(\varnothing) = 1$. $\Pi_i$ is the set of reduced-normal-form plans of player $i$, $\Pi_i(I_i)$ is the set of reduced-normal-form plans $\pi_i \in \Pi_i$ with that $\pi_i(\sigma_i(I_i)) = 1$, $\Pi_i(I_i, a_i)$ is the set of reduced-normal-form plans in $\Pi_i(I_i)$ such that $a_i \in A(I_i)$ is played in $\pi_i$. Then $\Pi_i(\varnothing) = \Pi_i$, and $\Pi_i(z)$ for each $z \in Z$ is the set of reduced-normal-form plans $\pi_i \in \Pi_i$ with that $\pi_i(\sigma_i(z)) = 1$. $\Delta(\Pi_i)$ is the set of mixed normal-form strategies, i.e., a probability distribution over $\Pi_i$. $\boldsymbol{\Pi}_T = \times_{i \in T} \Pi_i$ is the set of pure coordinated strategies of the team, and $\mu_T \in \Delta(\boldsymbol{\Pi}_T)$ is a mixed coordinated strategy. $\boldsymbol{\Pi}_T(h) = \times_{i \in T} \Pi_i(h)$ for each $h \in H \cup Z$ is the set of pure coordinated strategies with that $\pi_i(\sigma_i(h)) = 1$ for each player $i \in T$, denoted by $\boldsymbol{\pi}_T(\boldsymbol{\sigma}_T(h)) = 1$ with $\boldsymbol{\pi}_T \in \boldsymbol{\Pi}_T$. For example, in Figure 1(a), the paths $a-b-d-1$ and $a-c-g-7$ show a $\boldsymbol{\pi}_T$ such that $\pi_i(\sigma_i(1)) = 1$ (node 1 is a leaf) for each player $i \in T$, i.e., $\boldsymbol{\pi}_T(\boldsymbol{\sigma}_T(1)) = 1$.

**Sequence-form strategies.** Given player $i$, a mixed sequence-form strategy is a vector $\boldsymbol{y}_i \in [0, 1]^{|\Sigma_i|}$, and a pure sequence-form strategy is a vector $\boldsymbol{y}_i \in \{0, 1\}^{|\Sigma_i|}$, which satisfy: $y_i(\varnothing) = 1$, and

$\sum_{a_i \in A(I_i)} y_i(I_i, a_i) = y_i(\sigma_i(I_i))$ for each $I_i \in \mathcal{I}_i$. The set of sequence-form strategies is $\mathcal{Y}_i$. Two strategies are equivalent if they assign the same probability for reaching each leaf node. Any pure (mixed) normal-form strategy of each player $i$ is equivalent to a pure (mixed) sequence-form strategy and vice versa (von Stengel, 1996). However, coordinated strategies in $\Delta(\mathbf{\Pi}_T)$ cannot be concisely represented by sequence-form strategies due to the imperfect recall of the team (Farina et al., 2018).

**TMECor.** A Team-Maxmin Equilibrium with Coordination device (TMECor) (Celli & Gatti, 2018) is a Nash equilibrium, where the opponent plays the strategy $\mu_o \in \Delta(\Pi_o)$ (equivalent to $\boldsymbol{y}_o \in \mathcal{Y}_o$), and the team plays $\mu_T \in \Delta(\mathbf{\Pi}_T)$. The team's expected utility over $(\mu_T, \boldsymbol{y}_o)$ is:

$$u_T(\mu_T, \boldsymbol{y}_o) = \sum_{z \in Z} \hat{u}_T(z) \boldsymbol{y}_o(\sigma_o(z)) \sum_{\boldsymbol{\pi}_T \in \mathbf{\Pi}_T(z)} \mu_T(\boldsymbol{\pi}_T)$$

where $\hat{u}_T(z) = u_T(z) p_c(z)$, and $p_c(z)$ is the chance probability of reaching $z$. For $(\boldsymbol{\pi}_T, \boldsymbol{y}_o)$,

$$u_T(\boldsymbol{\pi}_T, \boldsymbol{y}_o) = \sum_{z \in Z, \boldsymbol{\pi}_T \in \mathbf{\Pi}_T(z)} \hat{u}_T(z) \boldsymbol{y}_o(\sigma_o(z)).$$

An optimal solution TMECor can be found by solving the following optimization problem, which is equivalent to a linear program by dualizing the inner linear minimization problem over $\boldsymbol{y}_o$:

$$\max_{\mu_T \in \Delta(\mathbf{\Pi}_T)} \min_{\boldsymbol{y}_o \in \mathcal{Y}_o} u_T(\mu_T, \boldsymbol{y}_o). \tag{1}$$

**CG.** Normally, the algorithm Column Generation (CG) (Zhang et al., 2021; Farina et al., 2021; Zhang et al., 2022b) shown in Figure 1(c) is used to compute a TMECor because $|\Pi_i|$ is exponential in the size of the game tree (i.e., normal-form CG). CG starts from a restricted game $G'$, then solves $G'$ for the corresponding TMECor, and then uses the BRO (Best Response Oracle, i.e., solving the problem: $\max_{\boldsymbol{\pi}_T \in \mathbf{\Pi}_T} u_T(\boldsymbol{\pi}_T, \boldsymbol{y}_o)$) to compute a coordinated best response against the adversarial strategy $\boldsymbol{y}_o$ in the TMECor of $G'$. If this best response improves the team's utility obtained by the TMECor of $G'$, CG expands $G'$ by adding this best response to $G'$; otherwise, CG terminates. More details on variants of CG are shown in Appendix A.

**Team Belief DAG.** We introduce the definitions related to Team Belief Directed Acyclic Graph (TB-DAG) (Zhang et al., 2022b;c). When a piece of information is common knowledge to the team $T$ in an ATG, it is public to $T$. Two nodes $h$ and $h'$ within the same level (i.e., the same length of the paths from the root to both nodes) are indistinguishable (not public) to the team if there is an information set $I \in \mathcal{I}_T$ such that $h \preceq I$ and $h' \preceq I$. For example, in Figure 1(a), nodes $b$ and $c$ are indistinguishable to the team. A connected component of a graph induced by nodes' indistinguishable relation to the team is a public state of the team.

The TB-DAG is a decision problem of the team in an ATG. The nodes in $\mathcal{D}$ of the TB-DAG include a set of observation nodes $\mathcal{O}$ (e.g., rectangle nodes in Figure 1(b)), i.e., including the information observed by the team about states of the game, and a set of decision nodes called beliefs $\mathcal{B}$ (e.g., circle nodes in Figure 1(b)). Each belief or observation node includes a set of nodes in $H \cup Z$ that the team cannot distinguish based on their public information. Starting from the root $\{\varnothing\} \cup J^*$ as a decision node ($J^*$ is a set of nodes before reaching any node of the team, e.g., node $a$ in Figure 1(b)), the TB-DAG is constructed recursively, where beliefs alternate with observation nodes, as shown in Figure 1(b). Each edge outgoing from a belief $B$ is a joint action called prescription that assigns an action to each information set that shares some nodes with $B$. $\mathcal{A}(B) = \{\boldsymbol{a} \mid \boldsymbol{a} \in \times_{I \cap B \neq \varnothing} A(I)\}$ is the set of possible prescriptions at $B$. $B$ also includes a set $J$ of nodes that are on the path to the next decision node of the team but do not belong to the team. The observation node transiting from $B$ by taking $\boldsymbol{a}$ is: $B\boldsymbol{a} = \bigcup_{I \cap B \neq \varnothing, a_I \in \boldsymbol{a}} \{ha_I \mid h \in I \cap B\} \cup \{ha \mid h \in J, a \in A(h)\}$. For example, in Figure 1(b), circle node $a$ is the root with the unique prescription $\varnothing$ for the team ($\{\varnothing\}$ is omitted in node $a$), which will transit to an observation node $bc$ due to two actions of the opponent. The circle node $bc$ has four prescriptions because $bc$ includes two information sets $b$ and $c$, each of which has two actions. A belief is a leaf node if it contains a leaf node in the ATG, e.g., circle nodes labeled by numbers in Figure 1(b). An observation node $O$ is a set of nodes, which form a set of public states of the team (connected components of the graph induced by $O$), and each public state represents one action (outgoing edge) of $O$. For example, in Figure 1(b), observation node $fg$ (or $bc$) has only one action because its two nodes are connected and then only form one connected component, but observation node $ef$ has two actions because $ef$ includes two connected components, i.e., node $e$ and node $f$ are not connected.

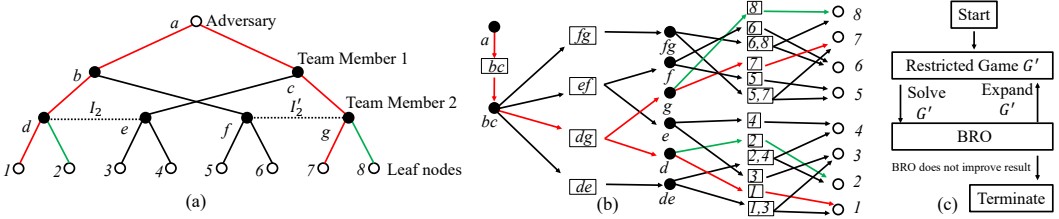

Figure 1: (a) An example of a game tree: paths $a$-$b$-$d$-1 and $a$-$c$-$g$-7 represent a pure coordinated strategy $\boldsymbol{\pi}_T^1$ for the team, and paths $a$-$b$-$d$-2 and $a$-$c$-$g$-8 represent $\boldsymbol{\pi}_T^2$ . (b) The TB-DAG for the example of (a): paths $a$-$bc$-$bc$-$dg$-$d$-1-1 and $a$-$bc$-$bc$-$dg$-$g$-7-7 represent a TB-DAG form strategy $\boldsymbol{x}^1$, and paths $a$-$bc$-$bc$-$dg$-$d$-2-2 and $a$-$bc$-$bc$-$dg$-$g$-8-8 represent $\boldsymbol{x}^2$. (c) The procedure of CG.

Let $\mathcal{E}$ be the set of edges in the TB-DAG of an EFG. The pseudocode for generating the TB-DAG is shown in Algorithm 1 of Appendix A.5. In a TB-DAG, a pure TB-DAG form strategy is $\boldsymbol{x} \in \{0,1\}^{|\mathcal{D}|}$, and a mixed TB-DAG form strategy is $\boldsymbol{x} \in [0,1]^{|\mathcal{D}|}$, which are constrained by: $x(B) = \sum_{\boldsymbol{a} \in \mathcal{A}(B)} x(B\boldsymbol{a})$ and $x(B) = \sum_{(O,B) \in \mathcal{E}} x(O)$ for $B \in \mathcal{B}$ with $x(\{\varnothing\} \cup J^*) = 1$, where $\{\varnothing\} \cup J^*$ represents the root. The set of the TB-DAG form strategies is equivalent to the set of the team's coordinated strategies in an EFG (Zhang et al., 2022c). By using $\boldsymbol{x}$ to replace $\mu_T$ in Eq.(1), we obtain a TMECor by solving Problem (1) through the TB-DAG.

## 3 DAG-BASED COLUMN GENERATION

Solving Problem (1) is challenging because the team's coordinated strategy space is exponential in the size of the game tree. With the procedure shown in Figure 1(c), CG can mitigate this challenge but still converges slowly in large games due to such a large strategy space. It has been shown that the TB-DAG-based algorithms are more efficient than CG in games with a small TB-DAG (Zhang et al., 2022b;c). However, solving Problem (1) through the TB-DAG is impractical in large games because the size of the TB-DAG is still exponential in the size of the game tree (Zhang et al., 2022c). To speed up, one straightforward idea is applying CG with the procedure shown in Figure 1(c) to the TB-DAG, which also is the direct application of the sequence-form double oracle (Bosansky et al., 2014) to the TB-DAG. That is, we compute a TMECor with a limited size of the TB-DAG, and then expand the DAG by computing a best response over the whole TB-DAG form strategy space, i.e., solving the problem: $\max_{\boldsymbol{x}} u_T(\boldsymbol{x}, \boldsymbol{y}_o)$. However, it is inefficient to compute a best response over the whole exponential-sized TB-DAG. As shown in experiments, this exponential size will make CG with the DAG-based BRO very inefficient. More details on related work are shown in Appendix A.

### 3.1 A NOVEL CG ALGORITHM FRAMEWORK

To solve ATGs more efficiently, we propose our novel TB-DAG CG (**DCG**) framework based on two observations: 1) by exploiting the team's correlation property to solve the BRO's integer program significantly faster, the state-of-the-art BRO (Zhang et al., 2021; Farina et al., 2021) should significantly outperform the BRO computing a best response over the whole exponential-sized TB-DAG without such a correlation property available, and 2) the TB-DAG formed by a set of TB-DAG form strategies transformed from a set of coordinated strategies could represent new coordinated strategies, as shown in Example 1 and Theorem 4 of Appendix C.

*Example* 1. TB-DAG form strategies $\boldsymbol{x}^1$ and $\boldsymbol{x}^2$ in Figure 1(b) are equivalent to coordinated strategies $\boldsymbol{\pi}_T^1$ and $\boldsymbol{\pi}_T^2$ in Figure 1(a), respectively. Let $\mathcal{D}(\{\boldsymbol{\pi}_T^1, \boldsymbol{\pi}_T^2\})$ be the TB-DAG induced by $\{\boldsymbol{\pi}_T^1, \boldsymbol{\pi}_T^2\}$, i.e., consisting of beliefs and actions reachable in $\boldsymbol{x}^1$ and $\boldsymbol{x}^2$. Then the TB-DAG form strategy space of $\mathcal{D}(\{\boldsymbol{\pi}_T^1, \boldsymbol{\pi}_T^2\})$ includes all combinations of beliefs and actions reachable in $\boldsymbol{x}^1$ and $\boldsymbol{x}^2$. For example, a TB-DAG form strategy $\boldsymbol{x}^3$ with paths $a$-$bc$-$bc$-$dg$-$d$-1-1 and $a$-$bc$-$bc$-$dg$-$g$-8-8 (i.e., the probability of reaching nodes 1 and 8 is 1) in Figure 1(b) is obtained by using the beliefs and actions represented by the sub-path $g$-8-8 in $\boldsymbol{x}^2$ to replace the beliefs and actions represented by the sub-path $g$-7-7 in $\boldsymbol{x}^1$ to make $\boldsymbol{x}^1$ become $\boldsymbol{x}^3$. $\boldsymbol{x}^3$ is in the TB-DAG form strategy space of $\mathcal{D}(\{\boldsymbol{\pi}_T^1, \boldsymbol{\pi}_T^2\})$ and is equivalent to the coordinated strategy $\boldsymbol{\pi}_T^3$ with paths $a$-$b$-$d$-1 and $a$-$c$-$g$-8 in Figure 1(a). However, $\boldsymbol{\pi}_T^3 \notin \Delta\{\boldsymbol{\pi}_T^1, \boldsymbol{\pi}_T^2\}$ because only using $\boldsymbol{\pi}_T^1$ and $\boldsymbol{\pi}_T^2$ cannot guarantee the probability of reaching leaf nodes 1 and 8 is 1. That is, any combinations of $\boldsymbol{\pi}_T^1$ and $\boldsymbol{\pi}_T^2$ cannot represent $\boldsymbol{\pi}_T^3$.

As shown in Example 1, the TB-DAG formed by a set of TB-DAG form strategies transformed from a set of coordinated strategies could represent new coordinated strategies due to the new combinations of beliefs and actions in this TB-DAG. Then this TB-DAG form strategies will represent more new coordinated strategies in larger games with wider or deeper game trees (having more combinations of beliefs and actions). It means that, if we expand the restricted game $G'$ in CG by using the TB-DAG form strategies instead of coordinated strategies, the corresponding CG could converge in fewer iterations in large games. Based on the above observations, with the similar procedure shown in Figure 1(c), the procedures of our DCG are: 1) computing a coordinated best response (e.g., through the existing BRO in Zhang et al. (2021); Farina et al. (2021)), 2) transforming this coordinated best response into the TB-DAG form, and then 3) adding this equivalent TB-DAG form strategy to the restricted game. The procedure of transforming a coordinated strategy into the TB-DAG form is similar to the procedure for generating the whole TB-DAG mentioned in the previous section, except that we only consider actions played by the team in the best response strategy with nonzero probabilities. The details of this procedure are shown in Appendix A.6. Intuitively, DCG should perform worse than the normal-form CG because DCG has an extra step for the transformation at each iteration. However, this intuition is contradicted by that DCG can converge in significantly fewer iterations than the normal-form CG and then outperforms the normal-form CG when the benefit from reducing the number of interactions for convergence surpasses the cost of the transformation.

The normal-form CG with the BRO based on a semi-randomized coordinated strategy was shown (Zhang et al., 2021) to converge to a TMECor in at most $2^{|\Pi_1|\frac{|\Pi_T|}{\Pi_1}}$ iterations. Now we theoretically show that our DCG converges exponentially faster than the normal-form CG. This result is based on the size of the TB-DAG (Zhang et al., 2022c): The TB-DAG has at most $O^*((b(p+1))^w)$ edges, where $b$ is the branching factor, $p$ is the largest effective size (the number of distinct team sequences) of any public state (i.e., connected component), $w$ is the maximum number of information sets involved in any belief, and $O^*$ hides factors polynomial in the size of the original game tree (Zhang et al., 2022c). (All proofs are in Appendix C)

**Theorem 1.** *DCG with any BRO converges to a TMECor in at most $O^*((b(p+1))^w)$ iterations.*

$b$, $p$, and $w$ are exponentially smaller than the size of the game tree, but $|\Pi_i|$ is exponential in the size of the game tree. Therefore, theoretically, DCG converges exponentially faster than the normal-form CG.

## 3.2 EXISTING BRO FOR DCG AND THE CORRESPONDING TRANSFORMATION COST

DCG can employ any existing BRO to compute a coordinated best response in the original game tree against the adversarial strategy $y_o$ in the TMECor of the restricted game $G'$. There are two kinds of BRO: randomized BRO and pure BRO. The randomized BRO is the state-of-the-art BRO (Zhang et al., 2021; Farina et al., 2021), which computes a semi-randomized coordinated strategy in which one team member plays a randomized strategy. The pure BRO in Celli & Gatti (2018) computes a coordinated best response, a coordinated strategy, with a pure strategy for each team member. The difference between the randomized and pure strategies makes the randomized BRO and the pure BRO have different transformation costs for transforming a coordinated strategy into the TB-DAG form.

The transformation cost for transforming a coordinated strategy into the TB-DAG form is determined by the size of this transformed TB-DAG. In the previous section, we know that the size of a TB-DAG is influenced by the branching factor $b$. For the transformation for the pure BRO, $b$ is 1 due to the pure strategy. However, $b$ is usually larger than 1 in the randomized BRO due to the randomized strategy, so the randomized BRO has a higher transformation cost than the pure BRO. For example, the TB-DAG induced by the pure coordinated strategy $\pi_T^1$ in Figure 1(a) is very small and only involves the node $\boxed{dg}$ and its succeeding nodes in the paths of $x^1$ in Figure 1(b). However, if team member 1 plays a randomized strategy at information sets $b$ and $c$ of Figure 1(a), the size of the TB-DAG induced by this coordinated strategy increases about threefold, i.e., four nodes $\boxed{fg}$, $\boxed{ef}$, $\boxed{dg}$, $\boxed{de}$ and half of their succeeding nodes in Figure 1(b) are involved. Actually, the transformation cost for a pure coordinated best response is only polynomial in the size of the original game tree.

**Theorem 2.** *The size of the transformed TB-DAG for a pure coordinated best response is at most $O(|H \cup Z|)$.*

Therefore, pure BRO will give us a small transformation cost. Unfortunately, the existing pure BRO in Celli & Gatti (2018) is extremely inefficient (Zhang et al., 2021; Farina et al., 2021) (see details in Appendix A). Therefore, we will develop a more suitable BRO (i.e., efficient pure BRO) for DCG.

## 3.3 More Suitable BRO for DCG

To improve the scalability of DCG, we propose a more suitable BRO, i.e., our pure BRO. Specifically, to effectively represent the space of pure coordinated strategies, we extend the two-player von Stengel-Forges polytope (Von Stengel & Forges, 2008) to cases with multiple players playing pure strategies, i.e., a correlation plan $\boldsymbol{\xi} \in [0,1]^{|\boldsymbol{\Sigma}_T^{\bowtie}|}$ with $-i = T \setminus \{i\}$ and empty joint sequences $\boldsymbol{\varnothing}_{-i} = \times_{j \in T \setminus \{i\}} \varnothing$ and $\boldsymbol{\varnothing}_T = \times_{j \in T} \varnothing$ satisfies the following polynomial-sized set of constraints of the probability flow representing the team's correlation property: $\xi(\boldsymbol{\varnothing}_T) = 1$, and for each $\sigma_i \in \Sigma_i$, $\xi(\sigma_i, \boldsymbol{\varnothing}_{-i}) \in \{0,1\}$,

$$\sum_{a_i \in A(I_i)} \xi(\boldsymbol{\sigma}_{-i}, (I_i, a_i)) = \xi(\boldsymbol{\sigma}_{-i}, \sigma_i(I_i)) \quad \forall I_i \bowtie \boldsymbol{\sigma}_{-i}, I_i \in \mathcal{I}_i, \boldsymbol{\sigma}_{-i} \in \boldsymbol{\Sigma}_{-i}^{\bowtie}, i \in T \tag{2a}$$

$$\sum_{j \in T} \xi(\boldsymbol{\sigma}_T[j], \boldsymbol{\varnothing}_{-j}) + 1 - |T| \leq \xi(\boldsymbol{\sigma}_T) \leq \xi(\boldsymbol{\sigma}_T[i], \boldsymbol{\varnothing}_{-i}) \quad \forall \boldsymbol{\sigma}_T \in \boldsymbol{\Sigma}_T^{\bowtie}, i \in T. \tag{2b}$$

$\boldsymbol{y}_i \in \{0,1\}^{|\Sigma_i|}$ such that $y_i(\sigma_i) = \xi(\sigma_i, \boldsymbol{\varnothing}_{-i})$ for each $\sigma_i \in \Sigma_i$ represents a pure reduced-normal-form plan, and then $\xi(\boldsymbol{\sigma}_T(z))$ for $z \in Z$ represents the reaching probability of a team's pure coordinated strategy (see Lemmas 1 and 2 in Appendix C). Then we can obtain a pure BRO:

$$\max_{\boldsymbol{\xi}} \sum_{z \in Z} \hat{u}_T(z) \xi(\boldsymbol{\sigma}_T(z)) \boldsymbol{y}_o(\sigma_o(z)) \tag{3a}$$

$$\text{subject to } Eq.(2), \xi(\boldsymbol{\sigma}_T) \in [0,1] \quad \boldsymbol{\sigma}_T \in \boldsymbol{\Sigma}_T^{\bowtie}. \tag{3b}$$

**Theorem 3.** *The optimal solution $\boldsymbol{\xi}^*$ of Program (3) defines a pure best response against $\boldsymbol{y}_o$.*

Program (3) is our pure BRO for DCG, and Theorem 1 for convergence still holds with this BRO.

**Corollary 4.** *DCG with the pure BRO converges to a TMECor in at most $O^*((b(p+1))^w)$ iterations.*

## 4 Experimental Evaluation

We computationally evaluate the performance of DCG. We run all experiments on a machine with a 4-core 2.3GHz CPU (8 threads) and 16GB of RAM available by using CPLEX 20.1.

**Algorithms.** We denote the DCG with our pure BRO of solving Program (3) in Section 3.3 by DCG$_{\text{pure}}$. We consider four variants of DCG$_{\text{pure}}$: 1) CG$_{\text{pure}}$: normal-form CG with our pure BRO; 2) DCG$_{\text{random}}$: DCG with the BRO in Zhang et al. (2021); Farina et al. (2021) computing a semi-randomized coordinated strategy; and 3) DCG$_{\text{2random}}$: DCG with two-sided CG (Zhang et al., 2022b), i.e., computing two best-response semi-randomized strategies at each iteration, and each one corresponds to one player playing a randomized strategy. We consider two state-of-the-art normal-form CG algorithms: 1) CG$_{\text{random}}$: CG in Zhang et al. (2021); Farina et al. (2021) and 2) CG$_{\text{2random}}$: two-sided CG in Zhang et al. (2022b) computing two best-response semi-randomized strategies and transforming the sequence-form (randomized) strategy in each strategy into variables to be re-optimized, i.e., CG$_{\text{2random}}$ adds not only two best response strategies but also $|\Sigma_i|$ variables for the sequence-form strategy of each $i \in T$ with the corresponding constraints. For each of these algorithms, we consider one more variant: at each iteration, it solves the linear relaxation of the BRO first to see if it can output the optimal solution added to the restricted game; if it cannot do that, it solves the original mixed-integer BRO and adds all feasible solutions for the BRO from the CPLEX solution pool to the original game. These variants are DCG$_{\text{pure}}^{\text{linrelax}}$, DCG$_{\text{random}}^{\text{linrelax}}$, DCG$_{\text{2random}}^{\text{linrelax}}$, CG$_{\text{pure}}^{\text{linrelax}}$, CG$_{\text{random}}^{\text{linrelax}}$, and CG$_{\text{2random}}^{\text{linrelax}}$ for DCG$_{\text{pure}}$, DCG$_{\text{random}}$, DCG$_{\text{2random}}$, CG$_{\text{pure}}$, CG$_{\text{random}}$, and CG$_{\text{2random}}$, respectively. For these CG-based algorithms, we randomly initialize the restricted game with a coordinated strategy for the team, and for the CG-based algorithms with semi-randomized strategies, we initialize a uniform strategy for the corresponding player. We consider two additional baselines: 1) DAG: it directly solves the linear program for a TMECor after generating the whole TB-DAG (see Appendix A.5); and 2) CGdag: CG with the sequence-form BRO (Bosansky et al., 2014) on the whole TB-DAG.

**Game instances.** There are many EFGs available for experiments, but we only need EFGs with different widths and depths to verify that DCG performs better in games with wider or deeper game

| Game | Relatively Shallow Game Trees | | | | Deeper and Deeper Game Trees | | | | Relatively Deep Game Trees | | | | Iterations | |
|---|---|---|---|---|---|---|---|---|---|---|---|---|---|---|
| | $^3K^1 5$ | $^3K^1 6$ | $^3K^1 10$ | $^3K^1 13$ | $^3K^1 8$ | $^3K^2 8$ | $^3K^3 8$ | $^3K^4 8$ | $^3K^4 5$ | $^3K^4 6$ | $^3K^3 10$ | $^3K^2 13$ | $^3K^3 8$ | $^3K^2 13$ |
| $\Delta U$ | 6 | 6 | 6 | 6 | 6 | 9 | 12 | 15 | 15 | 15 | 12 | 9 | 12 | 9 |
| $|\Sigma_i|$ | 41 | 49 | 81 | 105 | 65 | 201 | 497 | 1113 | 696 | 835 | 621 | 326 | 497 | 326 |
| $|Z|$ | 780 | 1560 | 9360 | 22308 | 4368 | 14448 | 36624 | 82992 | 14820 | 29640 | 78480 | 73788 | 36624 | 73788 |
| Rank | 5 | 6 | 10 | 13 | 8 | 8 | 8 | 8 | 5 | 6 | 10 | 13 | 8 | 13 |
| Depth | 6 | 6 | 6 | 6 | 6 | 8 | 8 | 9 | 12 | 12 | 9 | 8 | 9 | 8 |
| Value | -0.025 | -0.024 | -0.016 | -0.012 | -0.019 | -0.008 | 0.007 | 0.016 | -0.014 | 0.006 | 0.011 | 0.0004 | 0.007 | 0.0004 |
| $DCG_{pure}$ | 0.86s | 2.2s | 33s | 295s | 11s | 52s | 203s | **642s** | 49s | 107s | **936s** | **21m** | 166 | 239 |
| $DCG_{pure}^{linrelax}$ | 0.92s | 2.5s | 48s | 374s | 17s | 61s | **180s** | 17m | 47s | **103s** | 19m | 33m | 63 | 117 |
| $DCG_{random}$ | 2.1s | 11s | >10h | >10h | 515s | 34m | 576m | >10h | 50s | 301s | >10h | >10h | 157 | - |
| $DCG_{random}^{linrelax}$ | 2.1s | 11s | >10h | >10h | 482s | 31m | 528m | >10h | **31s** | 257s | >10h | >10h | 66 | - |
| $DCG_{2random}$ | 3.2s | 19s | >10h | >10h | 861s | 579m | >10h | >10h | 97s | 595s | >10h | >10h | - | - |
| $DCG_{2random}^{linrelax}$ | 3.7s | 21s | >10h | >10h | 925s | 582m | >10h | >10h | 68s | 581s | >10h | >10h | - | - |
| $CG_{pure}$ | 0.56s | 0.96s | 5s | **12s** | 2.3s | 161s | 37m | >10h | 893s | 32m | 65m | 63m | 1379 | 1129 |
| $CG_{pure}^{linrelax}$ | 0.52s | 0.94s | 5.7s | 22s | 3.5s | 211s | 25m | 10h | 819s | 33m | 37m | 47m | 449 | 513 |
| $CG_{random}$ | 0.46s | 1s | 4s | 18s | 3s | 135s | 44m | >10h | 34m | 37m | 60m | 53m | 1482 | 1158 |
| $CG_{random}^{linrelax}$ | 0.48s | 0.98s | 6.2s | 23s | 2.7s | 210s | 20m | 337m | 630s | 18m | 39m | 65m | 411 | 542 |
| $CG_{2random}$ | **0.23s** | **0.61s** | **2.5s** | 15s | **1.3s** | **51s** | 272s | $\infty$ | 60s | 138s | $\infty$ | $\infty$ | 47 | - |
| $CG_{2random}^{linrelax}$ | 0.3s | 0.99s | 13s | 64s | 1.7s | 68s | $\infty$ | $\infty$ | 95s | $\infty$ | $\infty$ | $\infty$ | - | - |
| DAG | 0.28s | 2s | $\infty$ | $\infty$ | $\infty$ | $\infty$ | $\infty$ | $\infty$ | 91s | $\infty$ | $\infty$ | $\infty$ | - | - |
| CGdag | 35m | >10h | >10h | >10h | >10h | >10h | >10h | >10h | >10h | >10h | >10h | >10h | - | - |

Table 1: Results on Kuhn poker: $\infty$ means 'out of memory'.

| Game | Relatively Deep Game Trees | | | | | | | Iterations for Convergence | | | | | | |
|---|---|---|---|---|---|---|---|---|---|---|---|---|---|---|
| | $^3_1L^1_1 3$ | $^3_1L^1_1 4$ | $^3_3L^1_1 5$ | $^3_1L^1_1 6$ | $^3_1L^1_1 7$ | $^3_1L^2_0 10$ | $^3_1L^2_0 13$ | $^3_1L^1_1 3$ | $^3_1L^1_1 4$ | $^3_3L^1_1 5$ | $^3_1L^1_1 6$ | $^3_1L^1_1 7$ | $^3_1L^2_0 10$ | $^3_1L^2_0 13$ |
| $\Delta U$ | 21 | 21 | 21 | 21 | 21 | 15 | 15 | 21 | 21 | 21 | 21 | 21 | 15 | 15 |
| $|\Sigma_i|$ | 457 | 801 | 1241 | 1489 | 2073 | 2411 | 4070 | 457 | 801 | 1241 | 1489 | 2073 | 2411 | 4070 |
| $|Z|$ | 6477 | 20856 | 51215 | 29880 | 69510 | 164880 | 552551 | 6477 | 20856 | 51215 | 29880 | 69510 | 164880 | 552551 |
| Rank | 3 | 4 | 5 | 6 | 7 | 10 | 13 | 3 | 4 | 5 | 6 | 7 | 10 | 13 |
| Depth | 12 | 12 | 12 | 12 | 12 | 11 | 11 | 12 | 12 | 12 | 12 | 12 | 11 | 11 |
| Value | 0.215 | 0.107 | 0.025 | -0.015 | -0.035 | -0.031 | -0.025 | 0.215 | 0.107 | 0.025 | -0.015 | -0.035 | -0.031 | -0.025 |
| $DCG_{pure}$ | 40s | 381s | 117m | 357s | 103m | 725s | 119m | 82 | 130 | 239 | 110 | 246 | 167 | 264 |
| $DCG_{pure}^{linrelax}$ | 34s | 188s | **36m** | **324s** | **37m** | 21m | 238m | 32 | 42 | 60 | 52 | 63 | 94 | 119 |
| $DCG_{random}$ | 46s | 537s | 99m | 31m | 289m | >10h | >10h | 82 | 146 | 211 | 84 | 230 | - | - |
| $DCG_{random}^{linrelax}$ | 29s | 191s | 42m | 647s | 74m | >10h | >10h | 27 | 45 | 62 | 32 | 60 | - | - |
| $DCG_{2random}$ | 88s | 76m | 21m | 303m | >10h | >10h | >10h | 81 | 134 | - | 63 | 166 | - | - |
| $DCG_{2random}^{linrelax}$ | 36s | 723s | 173m | 524s | 66m | >10h | >10h | 20 | 31 | 40 | 22 | 42 | - | - |
| $CG_{pure}$ | 822s | 434m | >10h | >10h | >10h | 49m | >10h | 1149 | 4364 | - | - | - | 1172 | - |
| $CG_{pure}^{linrelax}$ | 549s | 151s | >10h | >10h | >10h | 28m | $\infty$ | 384 | 798 | - | - | - | 342 | - |
| $CG_{random}$ | 933s | 10h | >10h | >10h | >10h | 29m | >10h | 1132 | 4165 | - | - | - | 1020 | - |
| $CG_{random}^{linrelax}$ | 485s | 224m | >10h | >10h | >10h | 36m | $\infty$ | 326 | 816 | - | - | - | 411 | - |
| $CG_{2random}$ | 609s | $\infty$ | $\infty$ | 57m | $\infty$ | $\infty$ | $\infty$ | 73 | - | - | 59 | - | - | - |
| $CG_{2random}^{linrelax}$ | 308s | $\infty$ | $\infty$ | $\infty$ | $\infty$ | $\infty$ | $\infty$ | 33 | - | - | - | - | - | - |
| DAG | **0.75s** | **9s** | 43m | 24m | $\infty$ | $\infty$ | $\infty$ | - | - | - | - | - | - | - |
| CGdag | 30m | >10h | >10h | >10h | >10h | >10h | >10h | 88 | - | - | - | - | - | - |

Table 2: Results on Leduc Poker: $\infty$ means 'out of memory'.

trees. We then use two standard EFGs (Farina et al., 2018; 2021; Carminati et al., 2022): Kuhn poker and Leduc poker (details on them can be found in these references). $^nK^c r$: $n$-player Kuhn poker with $r$ ranks and at most $c$ bets. $^n_sL^{c_1}_{c_2} r$: $n$-player Leduc poker with $r$ ranks, at most $c_1$ bets in the first betting round, at most $c_2$ bets in the second betting round, and $s$ suits. We consider two dimensions of the game tree in each game: depth and width. A game with more bets or ranks is larger. That is, a game with more bets means that its game tree is deeper according to the maximum number of actions at any sequence of any team member. Similarly, a game with more ranks (proportional to the maximum number of information sets involved in any belief) means that its game tree is wider. As we discussed in Section 3, the size of the TB-DAG is mainly influenced by this maximum number of information sets involved in any belief, so the TB-DAG is larger in games with wider game trees (more ranks). In addition, the game tree in Leduc poker with two rounds is deeper than the game tree in Kuhn poker with only one round. A game tree is wide if the value of 'Rank' is relatively large, and a game tree is narrow if the value of 'Rank' is relatively small. Similarly, a game tree is deep if the value of 'Depth' is relatively large, and a game tree is shallow if the value of 'Depth' is relatively small. Without loss of generality, the last player is the opponent.

**Results.** Results in Tables 1 and 2 show the algorithms' performance on runtime for converging to a TMECor with target precision of the team value in a TMECor is $10^{-6}$ and the corresponding number of iterations for CG algorithms if they converge within 10 hours. Results on varying target precision values are shown in Appendix D. Results show that our proposed algorithm $DCG_{pure}$ and its relaxed version $DCG_{pure}^{linrelax}$ significantly outperform all baselines in large games with deep and wide game trees. Normal-form CG algorithms (i.e., $CG_{pure}$, $CG_{pure}^{linrelax}$, $CG_{random}$, $CG_{random}^{linrelax}$, $CG_{2random}$, and $CG_{2random}^{linrelax}$) are the fastest algorithms in games with shallow game trees, and the DAG-based linear

program (i.e., DAG) is the fastest algorithm in games with very narrow game trees (e.g., $_3^3\text{L}_1^13$) but runs out of memory in games with wide game trees. However, CGdag is not efficient in all games because it needs to compute a best response on the whole TB-DAG. Finally, $\text{DCG}_{\text{random}}$, $\text{DCG}_{\text{random}}^{\text{linrelax}}$, $\text{DCG}_{2\text{random}}$, and $\text{DCG}_{2\text{random}}^{\text{linrelax}}$ are relatively fast in games with narrow game trees but not efficient in games with wide game trees.

Our $\text{DCG}_{\text{pure}}$ is at least two orders of magnitude faster than prior state-of-the-art baselines in large games. $\text{DCG}_{\text{pure}}$ needs to transform the coordinated best response into the corresponding TB-DAG form at each iteration, so it runs relatively slower than normal-form CG algorithms in games with shallow game trees, as shown in the first column of Table 1. When the game tree grows deeper and deeper, our $\text{DCG}_{\text{pure}}$ performs closer and closer to normal-form CG algorithms first and then outperforms normal-form CG algorithms with larger and larger gaps, as shown in the second column of Table 1. Then, in games with relatively deep game trees, our $\text{DCG}_{\text{pure}}$ is at least two orders of magnitude faster than normal-form CG algorithms, as shown in the third column of Table 1 and the first column of Table 2. The relaxed version $\text{DCG}_{\text{pure}}^{\text{linrelax}}$ of $\text{DCG}_{\text{pure}}$, in most games with relatively narrow game trees, outperforms $\text{DCG}_{\text{pure}}$ because it could reduce the cost of calling the mixed-integer BRO. However, in games with relatively wide game trees, $\text{DCG}_{\text{pure}}$ outperforms $\text{DCG}_{\text{pure}}^{\text{linrelax}}$ because $\text{DCG}_{\text{pure}}^{\text{linrelax}}$ transforms too many coordinated strategies into the TB-DAG form, which incurs a very high cost in games with relatively wide game trees. Our $\text{CG}_{\text{pure}}$ is comparable with the prior state-of-the-art single-side $\text{CG}_{\text{random}}$, and its relaxed version $\text{CG}_{\text{pure}}^{\text{linrelax}}$ always outperforms $\text{CG}_{\text{pure}}$ because it could reduce the cost of calling the mixed-integer BRO. In large games (e.g., $_1^3\text{L}_0^213$), due to transforming too many coordinated strategies into the TB-DAG form, $\text{CG}_{\text{pure}}^{\text{linrelax}}$ could run out of memory. Results of $\text{DCG}_{\text{random}}$, $\text{DCG}_{\text{random}}^{\text{linrelax}}$, $\text{CG}_{\text{random}}$, and $\text{CG}_{\text{random}}^{\text{linrelax}}$ have the similar pattern.

$\text{DCG}_{2\text{random}}$ still performs worse than $\text{DCG}_{\text{pure}}$ and $\text{DCG}_{\text{random}}$ in these games because this two-sided CG-based algorithm transforms two strategies instead of one strategy into the TB-DAG form at each iteration. Similar to $\text{DCG}_{\text{random}}$ and $\text{DCG}_{\text{random}}^{\text{linrelax}}$, $\text{DCG}_{2\text{random}}$ and $\text{DCG}_{2\text{random}}^{\text{linrelax}}$ do not perform well in games with relatively wide game trees due to the extra cost of the transformation for randomized strategies. $\text{CG}_{2\text{random}}$ performs the best in small games with relatively shallow game trees, as shown in the first column of Table 1, but performs worse and worse in games with deeper and deeper game trees, as shown in the second column of Table 1. Overall, $\text{CG}_{2\text{random}}$ cannot perform well in large games because it adds too many variables and constraints to the program for solving the restricted game at each iteration and then usually runs out of memory. In addition, $\text{CG}_{2\text{random}}^{\text{linrelax}}$ generally performs worse than $\text{CG}_{2\text{random}}$ because $\text{CG}_{2\text{random}}^{\text{linrelax}}$ usually adds more variables and constraints than $\text{CG}_{2\text{random}}$.

Results on the number of iterations for convergence in Tables 1 and 2 further confirm our analysis. We can see that our DCG algorithms require significantly fewer iterations for convergence than $\text{CG}_{\text{pure}}$, $\text{CG}_{\text{pure}}^{\text{linrelax}}$, $\text{CG}_{\text{random}}$, and $\text{CG}_{\text{random}}^{\text{linrelax}}$. $\text{CG}_{2\text{random}}$ and its variant $\text{CG}_{2\text{random}}^{\text{linrelax}}$ require relatively few iterations for convergence in relatively small games, but they run out of memory in large games.

**Limitations.** Runtime values reported in this paper for baselines may be different from the runtime values reported in previous papers (Zhang et al., 2021; Farina et al., 2021; Zhang et al., 2022c;b) because results reported in different papers may be obtained from different settings (see details in Appendix E). Thus, to have a fair comparison with the previous baselines, all algorithms in our experiments are tested with the same setting.

## 5 CONCLUSIONS

In this paper, we develop a novel TB-DAG CG framework to compute a TMECor in an ATG by computing a coordinated best response in the original game first and then transforming it into the TB-DAG form. We further reduce the cost of transformation, which is based on a more suitable BRO for DCG to compute a coordinated best response with a pure strategy for each team member. We theoretically show that our algorithm converges exponentially faster than the state-of-the-art CG algorithms, and experimental results show that our algorithm is significantly faster than the state-of-the-art baselines and solves games that were previously unsolvable. In the future, by applying the multiagent learning framework PSRO (Lanctot et al., 2017) with the aid of deep learning techniques for the transformation step and the best response oracle, we believe that our algorithm framework can scale to very large-scale games. Thus, this paper creates a fundamental theory for applying the multiagent learning framework to compute optimal solutions in multiagent systems.

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

# Appendix

## A  RELATED WORK

Our DCG does not just directly apply CG to the TB-DAG. Our work provides the first efficient TB-DAG CG algorithm, which includes a novel CG algorithm framework and solves many games that were previously unsolvable. Our DCG algorithm computes a pure coordinated best response (i.e., a coordinated strategy represented by a joint normal-form strategy of the team) in the original game tree first and then transforms it into the TB-DAG form. Our DCG has an extra transformation step, which does not exist in previous CG algorithms. Our DCG significantly overcomes the limitation of previous CG algorithms for a TMECor with a novel CG algorithm framework and a suitable best response oracle.

1. **Novel algorithm framework**: In previous CG algorithms, the strategy representations in the restricted game and the best response oracle are the same. For example, the normal-form CG (Celli & Gatti, 2018; Farina et al., 2018; Zhang et al., 2021; Farina et al., 2021; Zhang et al., 2022b) (variants of normal-form double-oracle (McMahan et al., 2003) by using a single oracle) solves the restricted game with the team's normal-form strategy space (i.e., the team's coordinated strategy space) and then iteratively expands the team's coordinated strategy space via a normal-form best response oracle; and the sequence-form double oracle (Bosansky et al., 2014) solves the restricted game with the sequence-form strategy space and then iteratively expands the team's sequence-form strategy space via a sequence-form best response oracle. In contrast, our DCG adopts different strategy representations in the restricted game and the best response oracle: the strategy representation in the restricted game is sequence-form via the TB-DAG, and the strategy representation is normal-form in the best response oracle. These strategy representations are connected by our extra transformation procedure. Applying the framework of previous CG algorithms to compute a TMECor is challenging, and our new algorithm framework aims to overcome these challenges:

   (a) The challenge of normal-form CG (Celli & Gatti, 2018; Farina et al., 2018; Zhang et al., 2021; Farina et al., 2021; Zhang et al., 2022b): The normal-form CG solves the restricted game with the team's coordinated space and then iteratively expands the team's coordinated strategy space via a normal-form best response oracle. Normal-form CG converges slowly in large games due to the exponential-sized team's coordinated strategy space. Our DCG can overcome the challenge of normal-form CG because the TB-DAG form strategy space is significantly smaller than the team's coordinated strategy space. Intuitively, our DCG should not outperform the normal-form CG because our DCG needs an extra step for the transformation at each iteration. We contradict this intuition by showing that the TB-DAG formed by a set of TB-DAG form strategies transformed from a set of the team's normal-form strategies could represent new coordinated strategies of the team due to the new combinations of states and actions in this TB-DAG. This property makes our DCG converge in significantly fewer iterations than the normal-form CG in large games. Then DCG outperforms the normal-form CG when the benefit from reducing the number of interactions for convergence surpasses the cost of the transformation. We theoretically show that our DCG converges exponentially faster than the normal-form CG.

   (b) The challenge of sequence-from CG: The sequence-from CG directly applies CG to the TB-DAG (a sequence-form structure). That is, we can compute a TMECor with a limited size of the TB-DAG for the restricted game and then expand the TB-DAG by computing a best response over the whole TB-DAG of the original game. Unfortunately, it is inefficient to compute a best response over the whole exponential-sized TB-DAG. Therefore, the CG directly applied to the TB-DAG is inefficient in large games. Our DCG can overcome the challenge of sequence-form CG because we compute the normal-form best response in the original game tree with the team's correlation property available and the original game tree is exponentially smaller than the whole TB-DAG.

2. **Suitable best response oracle for DCG**: If we directly use the prior state-of-the-art best response oracle (Zhang et al., 2021; Farina et al., 2021), the resulting DCG suffers from a

very high cost of the transformation in large games because the best response computed by the prior state-of-the-art BRO (Zhang et al., 2021; Farina et al., 2021) involves randomized strategies and then induces a large TB-DAG. To further improve the scalability, we propose a more suitable BRO for DCG to reduce the cost of the transformation. That is, we propose an efficient pure BRO to compute a coordinated best response with a pure strategy for each team member, which will make sure the corresponding TB-DAG form is small enough and then reduce the cost of the transformation. Our pure BRO extends the two-player von Stengel-Forges polytope (Von Stengel & Forges, 2008) to cases with multiple players playing pure strategies to represent the space of pure coordinated strategies effectively. Our pure BRO is fundamentally different from the pure BRO in Celli & Gatti (2018), which expresses whether or not a leaf is reached by a pure joint normal-form strategy of all team members with $|Z|$ (the number of leaf nodes) integer variables. Our pure BRO expresses whether a sequence is played by a pure joint normal-form strategy, which involves only $\sum_{i \in T} |\Sigma_i|$ (the number of sequences of all players) integer variables. As shown in Tables 1 and 2, $\sum_{i \in T} |\Sigma_i|$ is significantly smaller than $|Z|$. The large number of integer variables makes the pure BRO in Celli & Gatti (2018) inefficient, which is the reason why the randomized BRO was developed in Zhang et al. (2021); Farina et al. (2021). Our pure BRO is different from the randomized BRO (Zhang et al., 2021; Farina et al., 2021) that exploits the two-player von Stengel-Forges polytope in two aspects:

(a) Their randomized BRO (Zhang et al., 2021; Farina et al., 2021) involves randomized strategies, but our pure BRO only includes pure strategies for team members to reduce the cost of the transformation, which also means that our pure BRO has more integer variables than the randomized BRO.

(b) Their randomized BRO (Farina et al., 2021) is only limited to the two-player case by directly exploiting the two-player von Stengel-Forges polytope, but our pure BRO can be applied to the cases with any number of players by extending the two-player von Stengel-Forges polytope to cases with multiple players.

Compared with other BRO algorithms, we understand that our pure BRO alone is not so novel, but our BRO is more suitable for DCG than other BRO algorithms. Indeed, experimental results show that DCG with our pure BRO significantly outperforms DCG with the randomized BRO, even though it has more integer variables than the randomized BRO. Our DCG with a novel CG algorithm framework (with an extra transformation step) and a pure BRO with more integer variables, as a whole, is novel.

Thus, this paper provides the first efficient TB-DAG CG algorithm and will be the base for applying the multiagent learning framework– Policy-Spaced Response Oracles (PSRO) (Lanctot et al., 2017) (a variant of CG) to the TB-DAG for a TMECor. Moreover, our TB-DAG CG with an extra transformation step and a pure BRO with more integer variables represents a new approach for computing a TMECor, and its surprising performance shows the promise of this approach and makes us understand TMECor better.

## A.1 RELATION BETWEEN DCG AND THE SEQUENCE-FORM DOUBLE ORACLE

Note that the sequence-form double oracle (Bosansky et al., 2014) is an extension of the normal-form double oracle (McMahan et al., 2003). Similar to other CG algorithms (i.e., single-oracle algorithms), as incremental strategy generation algorithms, our CG algorithm and the sequence-form double oracle (Bosansky et al., 2014) have a nearly-identical main loop: compute a best response (by any method), add the actions/sequences that are played in that best response to the strategy space, and repeat as necessary. Except for that common main loop, our algorithm is fundamentally different from the sequence-form double oracle (Bosansky et al., 2014):

1. We have different frameworks for strategy representations: The strategy representations in the restricted game and the best response oracle are the same in Bosansky et al. (2014). That is, the sequence-form double oracle (Bosansky et al., 2014) solves the restricted game with the sequence-form strategy space and then iteratively expands the team's sequence-form strategy space via a sequence-form best response oracle. In contrast, our DCG adopts different strategy representations in the restricted game and the best response oracle: the strategy representation in the restricted game is sequence-form via the TB-DAG, and

the strategy representation is normal-form in the best response oracle. These strategy representations are connected by our extra transformation procedure that transforms the normal-form strategy into the TB-DAG form. Based on the sequence-form double oracle (Bosansky et al., 2014), we can directly apply CG to the TB-DAG (a sequence-form structure). That is, we can compute a TMECor with a limited size of the TB-DAG for the restricted game and then expand the TB-DAG by computing a best response over the whole TB-DAG of the original game. Unfortunately, it is inefficient to compute a best response over the whole exponential-sized TB-DAG. Therefore, the CG directly applied to the TB-DAG is inefficient in large games. Our DCG can overcome the challenge of this sequence-form CG because we compute the normal-form (coordinated) best response in the original game tree with the team's correlation property available and the original game tree is exponentially smaller than the whole TB-DAG.

2. Our algorithm does not have the key features of the sequence-form double oracle (Bosansky et al., 2014) in the restricted game and the best response oracle.

   (a) The restricted game: In the sequence-form double oracle (Bosansky et al., 2014), the restricted game is represented by limited sequence-form strategies of two players. Then it needs to handle the following two primary complications "*that arise when we use sequences instead of full strategies in the double-oracle algorithm, both due to the fact that sequences do not necessarily define actions in all information sets: (1) a strategy computed in the restricted game may not be a complete strategy in the original game, because it does not define behavior for information sets that are not in the restricted game, and (2) it may not be possible to play every action from a sequence that is allowed in the restricted game, because playing a sequence can depend on having a compatible sequence of actions for the opponent*" (Bosansky et al., 2014, p. 840). However, in our DCG (a single-oracle algorithm), the restricted game includes all sequence-form strategies of the opponent and the limited team's TB-DAG form strategies generated by the equivalent (complete) normal-form strategies, which means that our DCG can avoid the above two primary complications of (Bosansky et al., 2014).

   (b) Best response oracle: The sequence-form double oracle (Bosansky et al., 2014) adopts the sequence-form best response oracle with the branch-and-bound approach, but our best response oracle computes a normal-form best response via solving a mixed-integer linear program.

3. Our DCG has an extra transformation step that transforms a best response from one strategy representation into another strategy representation, but the sequence-form double oracle (Bosansky et al., 2014) does not have such a transformation step.

## A.2 ABOUT THE VIEWPOINT OF APPLYING SEQUENCE-FORM DOUBLE ORACLE TO ATGS

Another viewpoint for presenting our algorithm is applying the sequence-form double oracle algorithm (Bosansky et al., 2014) to ATGs. Recall that the sequence-form double oracle algorithm in Bosansky et al. (2014) is only for two-player extensive-form perfect-recall zero-sum games. However, as we mentioned, an ATG is an imperfect-recall game. Thus, the algorithm in Bosansky et al. (2014) cannot be directly applied to original ATGs. To apply the algorithm in Bosansky et al. (2014) for solving ATGs, we need to transform the original strategy representation of ATGs into the TB-DAG form. Then, we can compute a TMECor with a limited size of the TB-DAG for the restricted game and then expand the TB-DAG by computing a best response over the whole TB-DAG of the original game. Unfortunately, it is inefficient to compute a best response over the whole exponential-sized TB-DAG. Therefore, the sequence-form double oracle algorithm in Bosansky et al. (2014) that is directly applied to the TB-DAG is inefficient in large games. To improve the scalability, similar to Section 3, we can propose a novel sequence-form oracle algorithm framework that includes 1) computing a coordinated best response, 2) transforming this coordinated best response into the TB-DAG form, and then 3) adding this equivalent TB-DAG form strategy to the restricted game. This algorithm framework has an extra transformation step and needs a suitable BRO to reduce the cost of the transformation.

We can see that the above algorithm procedure is the same as our DCG. We chose to present our algorithm from the viewpoint of applying CG to the TB-DAG because:

1. As we mentioned, both viewpoints will result in the same algorithm.

2. In the mainstream literature on ATGs (Celli & Gatti, 2018; Farina et al., 2018; Zhang et al., 2021; Farina et al., 2021; Zhang et al., 2022b), CG was used to solve ATGs because the team's strategy space is in normal form and is exponentially larger than the strategy space of the opponent (adversary).

3. Because the strategy space of the opponent (adversary) is exponentially smaller than the team's strategy space, we only expand the strategy space of the team. Thus, we do not need the key features of the sequence-form double oracle algorithm (Bosansky et al., 2014) in the restricted game (i.e., incomplete strategy and unallowed actions), as we mentioned in Appendix A.1.

That is, both viewpoints give us the same procedure and do not need key feature of the sequence-form double oracle algorithm in Bosansky et al. (2014), but using CG is the tradition in the literature on ATGs. Therefore, we choose the viewpoint of applying CG to the TB-DAG.

### A.3    About the Representation Size in BRO

Recall that the sequence-from CG directly applies CG to the TB-DAG (a sequence-form structure). That is, we can compute a TMECor with a limited size of the TB-DAG for the restricted game and then expand the TB-DAG by computing a best response over the whole TB-DAG of the original game. Unfortunately, it is inefficient to compute a best response over the whole exponential-sized TB-DAG. Therefore, the CG directly applied to the TB-DAG is inefficient in large games. Our DCG can overcome the challenge of sequence-form CG because we compute the normal-form best response in the original game tree with the team's correlation property available and the original game tree is exponentially smaller than the whole TB-DAG.

Our pure BRO ensures that only one joint action for each reachable belief will be used to expand the TB-DAG and then reduce the cost of the transformation. We exploit the team's correlation property defined in Eq.(2a) (a probability flow) to improve the scalability of our BRO. In the original two-player von Stengel-Forges polytope (Von Stengel & Forges, 2008) for this probability flow, the constraints involve all relevant sequences. If we consider all relevant joint sequences for multiple team members, the number of constraints in our BRO will be exponential in the size of the game tree. To reduce the number of constraints, we consider only a subset of relevant joint sequences for nodes and sequences of the original game tree (i.e., $\mathbf{\Sigma}_T^{\bowtie}$) and limit the relevant relation of an information set and a joint sequence to $\mathbf{\Sigma}_T^{\bowtie}$. That is, $\mathbf{\Sigma}_T^{\bowtie} = \{\boldsymbol{\sigma}_T(h_i) \mid h_i \in H \cup Z\} \cup \{(\sigma_i, \boldsymbol{\varnothing}_{-i}) \mid \sigma_i \in \Sigma_i, i \in T\}$ and $I_i \bowtie \boldsymbol{\sigma}_{-i}$ only if $(\sigma_i(I_i), \boldsymbol{\sigma}_{-i}) \in \mathbf{\Sigma}_T^{\bowtie}$. We further add constraints in Eq.(2b) to ensure that $\xi(\boldsymbol{\sigma}_T) = \prod_{i \in T} \xi(\boldsymbol{\sigma}_T[i], \boldsymbol{\varnothing}_{-i})$ for each $\boldsymbol{\sigma}_T \in \Sigma_T^{\bowtie}$. Therefore, our pure BRO shown in Program (3) has polynomial-sized constraints, i.e., $O(|H \cup Z||T|)$ constraints.

### A.4    About Column Generation for Sparser Solutions in Subgame Solving

Zhang et al. (2022a) provided a subgame technique to solve ATGs and proposed using CG for sparse solutions in the subgame solving algorithm. That is, for each reachable belief in each public state, they compute a best response starting from this belief. To keep the number of these reachable beliefs small, they create sparse blueprints in the subgame solving algorithm by using a CG algorithm because "the support size of the blueprint generated by a CG algorithm, scales linearly with the number of iterations, which, under reasonable time constraints, rarely exceeds the hundreds" (Zhang et al., 2022a). Their sparse blueprint and our pure best response are both sparse solutions, but theirs is different from our suitable BRO:

1. The goals are different: Our goal is to reduce the cost of the transformation in our DCG, but their goal is to reduce the number of times that the BRO is called.

2. The sparsity concepts are different: Their sparse blueprint is about a small support size of the blueprint (a mixed strategy of the gadget game), but our pure best response is just about an action for each reachable state/belief.

3. The approaches are different: We propose a new suitable BRO, i.e., pure BRO, to improve our DCG, but they just directly use an existing CG algorithm for a sparse blueprint, i.e., they do not provide a new BRO algorithm.

---

**Algorithm 1** Expanding the TB-DAG

---

1: **Function** ADDBELIEF($B, \mathcal{D}, br$):
2: **if** $B \notin \mathcal{D}$ **then**
3:    Add $B$ to $\mathcal{D}$
4:    **if** $B = \{z\}$ for $z \in Z$ **then**
5:       Make $B$ a leaf node and **Return** $B$
6:    **end if**
7:    $\mathcal{I}' \leftarrow \{I \cap B \neq \varnothing, I \in \mathcal{I}'\}$
8:    $J \leftarrow \{h \in B, \rho(h) \in \{o, c\}\}$, $\rho(h)$ is the player acting at node $h$
9:    $A_I \leftarrow \{a_i \in A(I) : br(I, a_i) > 0\}, \forall I \in \mathcal{I}'$
10:    **for** $\boldsymbol{a} \in \times_{I \in \mathcal{I}'} A_I$ **do**
11:       $B\boldsymbol{a} \leftarrow \cup_{I \in \mathcal{I}', a_I \in \boldsymbol{a}}\{ha_I \mid h \in I \cap B\} \cup \{ha \mid h \in J, a \in A(h)\}$
12:       add edge $B \to$ ADDOBSERVE($B\boldsymbol{a}, \mathcal{D}, br$)
13:    **end for**
14: **end if**
   **Function** ADDOBSERVE($O, \mathcal{D}, br$):
15: **if** $O \notin \mathcal{D}$ **then**
16:    Add $O$ to $\mathcal{D}$
17:    **for** each connected component $P$ for $O$ **do**
18:       add edge $O \to$ ADDBELIEF($P, \mathcal{D}, br$)
19:    **end for**
20: **end if**
   **Function** EXPANDTBDAG($\mathcal{D}, br$):
21: ADDBELIEF($\{\varnothing\} \cup J^*, \mathcal{D}, br$), where $J^*$ is a set of nodes before reaching any node of the team

---

## A.5   Generating the TB-DAG

The procedure for generating the TB-DAG is shown in Algorithm 1, which starts with EXPANDTBDAG($\varnothing, br$) at Line 21, where $br$ assigns 1 to each sequence $(I, a_i)$ of the team, i.e., $A_I = A(I)$ in Line 9. Note that we do not perform optimization tricks in Zhang et al. (2022c) after generating the TB-DAG. The reason is that: DAG, i.e., solving the linear program based on the whole TB-DAG, is fast enough in games with narrow game trees, but it will run out of memory in games with wide game trees because the TB-DAG is too large, where optimization tricks after generating the whole TB-DAG cannot mitigate the problem causing by the memory requirement much.

## A.6   Transforming a Coordinated Strategy into the TB-DAG Form

The procedure of transforming a coordinated strategy into the TB-DAG form is shown in Algorithm 1, where $br$ represents the best response of the team, i.e., a coordinated strategy, and $br(I, a_i)$ at Line 9 is the probability to play sequence $(I, a_i)$ according to $br$. This procedure is similar to the procedure for generating the whole TB-DAG mentioned in the previous section, except that, in Line 9 of Algorithm 1, we only consider actions played by the team in the best response strategy with nonzero probabilities.

# B   Construction

# C   Proofs

**Theorem 1.** *DCG with any BRO converges to a TMECor in at most $O^*((b(p + 1))^w)$ iterations.*

*Proof.* The TB-DAG has at most $O^*(b(p + 1))^w$ edges (Zhang et al., 2022c). In the worst case, DCG will add all of these edges to restricted game $G'$. Then DCG converges to a TMECor in at most $O^*(b(p + 1))^w$ iterations.     □

**Theorem 2.** *The size of the transformed TB-DAG for a pure coordinated best response is at most $O(|H \cup Z|)$.*

*Proof.* The size of this transformed TB-DAG for a pure coordinated best response is at most $O(|H \cup Z|)$ because:

1. The number of beliefs at any level in the TB-DAG form is not greater than the number of nodes in the corresponding level of the original game tree because each belief is a connected component in its parent (an observation node), and these beliefs in this transformed TB-DAG do not share nodes in the original game tree due to the unique prescription in each belief. Therefore, the number of beliefs in this transformed TB-DAG is less than $|H \cup Z|$.

2. Each observation node corresponds to one outgoing edge (a prescription) of a belief, and there is only one prescription for each belief now. Therefore, the number of observation nodes in this transformed TB-DAG is less than $|H|$.

$\square$

To show Theorem 3, we first show that the above correlation plan $\boldsymbol{\xi}$ defines a pure sequence-form strategy, i.e., a reduced-normal-form plan, for each player $i \in T$. Recall that $-i = T \setminus \{i\}$ and empty joint sequences $\boldsymbol{\varnothing}_{-i} = \times_{j \in T \setminus \{i\}} \varnothing$ and $\boldsymbol{\varnothing}_T = \times_{j \in T} \varnothing$.

**Lemma 1.** *Let $\boldsymbol{y}_i \in \{0,1\}^{|\Sigma_i|}$ such that $y_i(\sigma_i) = \xi(\sigma_i, \boldsymbol{\varnothing}_{-i})$ for each $\sigma_i \in \Sigma_i$, then $\boldsymbol{y}_i$ is a pure sequence-form strategy and also a reduced-normal-form plan.*

*Proof.* By Eq.(2), we have $y_i(\varnothing) = 1$, and $\sum_{a_i \in A(I_i)} y_i(I_i, a_i) = y_i(\sigma_i(I_i))$ for each $I_i \in \mathcal{I}_i$. Therefore, $\boldsymbol{y}_i$ is a pure sequence-form strategy. $y_i(I_i, a_i) = 1$ if $I_i$ is reachable and $a_i \in A(I_i)$ is played in $\pi_i$, which is the definition of a reduced-normal-form plan. Then $\boldsymbol{y}_i$ is a reduced-normal-form plan. $\square$

Now we show that the probability of each joint sequence in the correlation plan $\boldsymbol{\xi}$ is the product of the probabilities for playing the individual sequence of each team member.

**Lemma 2.** *For each $\boldsymbol{\sigma}_T \in \Sigma_T^{\bowtie}, \xi(\boldsymbol{\sigma}_T) = \prod_{i \in T} \xi(\boldsymbol{\sigma}_T[i], \boldsymbol{\varnothing}_{-i})$.*

*Proof.* For each $\boldsymbol{\sigma}_T \in \Sigma_T^{\bowtie}$ and each $i \in T$, $\xi(\boldsymbol{\sigma}_T[i], \boldsymbol{\varnothing}_{-i}) \in \{0,1\}$. By Eq.(2b), (1) if there is $i \in T$ such that $\xi(\boldsymbol{\sigma}_T[i], \boldsymbol{\varnothing}_{-i}) = 0$, then $\xi(\boldsymbol{\sigma}_T) = 0$; and (2) if for all $i \in T$ with $\xi(\boldsymbol{\sigma}_T[i], \boldsymbol{\varnothing}_{-i}) = 1$, then $\xi(\boldsymbol{\sigma}_T) = 1$. Therefore, for each $\boldsymbol{\sigma}_T \in \Sigma_T^{\bowtie}, \xi(\boldsymbol{\sigma}_T) = \prod_{i \in T} \xi(\boldsymbol{\sigma}_T[i], \boldsymbol{\varnothing}_{-i})$.

$\square$

**Theorem 3.** *The optimal solution $\boldsymbol{\xi}^*$ of Program (3) defines a pure best response against $\boldsymbol{y}_o$.*

*Proof.* By Lemmas 1 and 2, let $y_i(\sigma_i) = \xi(\sigma_i, \boldsymbol{\varnothing}_{-i})$ for each $i \in T$ and $\sigma_i \in \Sigma_i$, then $\xi(\boldsymbol{\sigma}_T(z)) = 1$ for each $z \in Z$ could represent that $z$ is reachable according to the coordinated strategy $\times_{i \in T} \boldsymbol{y}_i$, i.e., $\times_{i \in T} \boldsymbol{y}_i \in \boldsymbol{\Pi}_T(z)$. Then we can compute a pure best response against $\boldsymbol{y}_o$ via Program (3), i.e., the optimal solution $\boldsymbol{\xi}^*$ of Program (3) defines a pure best response against $\boldsymbol{y}_o$. $\square$

Let $\boldsymbol{\Pi}'_T \subseteq \boldsymbol{\Pi}_T$. $\mathcal{D}(\boldsymbol{\Pi}'_T)$ is the TB-DAG induced by $\boldsymbol{\Pi}'_T$ according to the equivalent TB-DAG form strategy of each coordinated strategy in $\boldsymbol{\Pi}'_T$. Let $\boldsymbol{\Pi}_T(\mathcal{D}(\boldsymbol{\Pi}'_T))$ be the set of coordinated strategies induced by $\mathcal{D}(\boldsymbol{\Pi}'_T)$ according to the equivalent coordinated strategy of each TB-DAG form strategy in $\mathcal{D}(\boldsymbol{\Pi}'_T)$. Inspired by Example 1, we have the following formal result.

**Theorem 4.** *Given $\boldsymbol{\pi}_T^1$ and $\boldsymbol{\pi}_T^2 \in \boldsymbol{\Pi}_T$, there is $\boldsymbol{\pi}_T^3 \in \boldsymbol{\Pi}_T(\mathcal{D}(\{\boldsymbol{\pi}_T^1, \boldsymbol{\pi}_T^2\}))$ such that $\boldsymbol{\pi}_T^3 \notin \Delta(\{\boldsymbol{\pi}_T^1, \boldsymbol{\pi}_T^2\})$ if:*

1. *There are two nodes in two different unconnected information sets, i.e., $h_1 \in I_1 \in \mathcal{I}_T$ and $h_2 \in I_2 \in \mathcal{I}_T$ with $I_1 \neq I_2$, $I_1 \not\rightleftharpoons I_2$, such that $h_1$ and $h_2$ are exclusively reachable in $I_1$ and $I_2$ by $\boldsymbol{\pi}_T^1$ and $\boldsymbol{\pi}_T^2$, respectively, i.e.,*

$$\boldsymbol{\pi}_T^1(\boldsymbol{\sigma}_T(h_1)) = \boldsymbol{\pi}_T^1(\boldsymbol{\sigma}_T(h_2)) = \boldsymbol{\pi}_T^2(\boldsymbol{\sigma}_T(h_1)) = \boldsymbol{\pi}_T^2(\boldsymbol{\sigma}_T(h_2)) = 1$$

*and for any $h_1' \in I_1$ with $h_1' \neq h_1$ and $h_2' \in I_2$ with $h_2' \neq h_2$*

$$\boldsymbol{\pi}_T^1(\boldsymbol{\sigma}_T(h_1')) = \boldsymbol{\pi}_T^1(\boldsymbol{\sigma}_T(h_2')) = \boldsymbol{\pi}_T^2(\boldsymbol{\sigma}_T(h_1')) = \boldsymbol{\pi}_T^2(\boldsymbol{\sigma}_T(h_2')) = 0.$$

| | Runtime: $^3_3L^1_13$ with $\Delta U = 21$ | | | | | | | |
|---|---|---|---|---|---|---|---|---|
| Target Precision | $0.1\times\Delta U$ | $0.05\times\Delta U$ | $0.01\times\Delta U$ | $0.005\times\Delta U$ | $0.001\times\Delta U$ | 0.01 | $10^{-4}$ | $10^{-6}$ |
| $\text{DCG}_{\text{pure}}$ | 1.7s | 3.3s | 10s | 13s | 28s | 32s | 40s | 40s |
| $\text{DCG}_{\text{pure}}^{\text{linrelax}}$ | 2s | 3s | 10s | 15s | 25s | 27s | 34s | 34s |
| $\text{DCG}_{\text{random}}$ | 1.3s | 4s | 12s | 14.5s | 31s | 38s | 45.5s | 46s |
| $\text{DCG}_{\text{random}}^{\text{linrelax}}$ | 2s | 4.8s | 10.4s | **11.5s** | **22s** | **25s** | **29s** | **29s** |
| $\text{DCG}_{\text{2random}}$ | 1.8s | 6.3s | 19s | 28s | 47s | 59s | 85.8s | 88s |
| $\text{DCG}_{\text{2random}}^{\text{linrelax}}$ | 4s | 5s | 14s | 16s | 26s | 32s | 36s | 36s |
| $\text{CG}_{\text{pure}}$ | 2.4s | 6.6s | 47s | 115s | 301s | 414s | 813s | 822s |
| $\text{CG}_{\text{pure}}^{\text{linrelax}}$ | 1.5s | 6.3s | 46s | 83s | 227s | 312s | 530s | 549s |
| $\text{CG}_{\text{random}}$ | 1s | 6s | 50s | 90s | 325s | 374s | 907s | 933s |
| $\text{CG}_{\text{random}}^{\text{linrelax}}$ | 0.8s | 7s | 43s | 77s | 223s | 276s | 435s | 485s |
| $\text{CG}_{\text{2random}}$ | **0.7s** | **2s** | **9s** | 14.6s | 92s | 148s | 578s | 609s |
| $\text{CG}_{\text{2random}}^{\text{linrelax}}$ | **0.7s** | 11.6s | 32s | 46s | 107s | 136s | 298s | 308s |
| | Iterations: $^3_3L^1_13$ with $\Delta U = 21$ | | | | | | | |
| Target Precision | $0.1\times\Delta U$ | $0.01\times\Delta U$ | $0.01\times\Delta U$ | $0.005\times\Delta U$ | $0.001\times\Delta U$ | 0.01 | $10^{-4}$ | $10^{-6}$ |
| $\text{DCG}_{\text{pure}}$ | 7 | 14 | 34 | 40 | 66 | 72 | 82 | 82 |
| $\text{DCG}_{\text{pure}}^{\text{linrelax}}$ | 3 | 4 | 13 | 17 | 25 | 27 | 32 | 32 |
| $\text{DCG}_{\text{random}}$ | 4 | 12 | 36 | 40 | 64 | 73 | 81 | 82 |
| $\text{DCG}_{\text{random}}^{\text{linrelax}}$ | 4 | 6 | 12 | 13 | 22 | 24 | 27 | 27 |
| $\text{DCG}_{\text{2random}}$ | 3 | 11 | 30 | 41 | 58 | 66 | 80 | 81 |
| $\text{DCG}_{\text{2random}}^{\text{linrelax}}$ | 3 | 5 | 11 | 12 | 16 | 18 | 20 | 20 |
| $\text{CG}_{\text{pure}}$ | 9 | 28 | 151 | 284 | 565 | 712 | 1140 | 1149 |
| $\text{CG}_{\text{pure}}^{\text{linrelax}}$ | 3 | 11 | 55 | 85 | 187 | 242 | 372 | 384 |
| $\text{CG}_{\text{random}}$ | 6 | 30 | 163 | 245 | 582 | 630 | 1111 | 1132 |
| $\text{CG}_{\text{random}}^{\text{linrelax}}$ | 5 | 17 | 62 | 85 | 175 | 204 | 297 | 326 |
| $\text{CG}_{\text{2random}}$ | 2 | 4 | 10 | 14 | 37 | 44 | 72 | 73 |
| $\text{CG}_{\text{2random}}^{\text{linrelax}}$ | 2 | 3 | 7 | 8 | 14 | 17 | 32 | 33 |

Table 3: Results on Leduc Poker $^3_3L^1_13$: $\infty$ means 'out of memory'.

2. $h_1$ and $h_2$ (i.e., $I_1$ and $I_2$) both have two different actions, i.e., $a_1, a'_1 \in A(h_1)$ with $a_1 \neq a'_1$ and $a_2, a'_2 \in A(h_2)$ with $a_2 \neq a'_2$, such that different actions are played by $\boldsymbol{\pi}^1_T$ and $\boldsymbol{\pi}^2_T$, i.e.,

$$\boldsymbol{\pi}^1_T(I_1, a_1) = \boldsymbol{\pi}^1_T(I_2, a_2) = \boldsymbol{\pi}^2_T(I_1, a'_1) = \boldsymbol{\pi}^2_T(I_2, a'_2) = 1.$$

*Proof.* We construct a new strategy $\boldsymbol{\pi}^3_T$, which is initialized by $\boldsymbol{\pi}^3_T = \boldsymbol{\pi}^1_T$. By the definition of $\boldsymbol{\pi}^1_T$, $h_1$ and $h_2$ are reachable in the current $\boldsymbol{\pi}^3_T$ that plays $a_1$ and $a_2$ in these nodes. To be different from $\boldsymbol{\pi}^1_T$ and $\boldsymbol{\pi}^2_T$, we modify the current $\boldsymbol{\pi}^3_T$ by: for each node $h' \in H_T \cup Z$ with $h_1 \preceq h'$, $\boldsymbol{\pi}^3_T(\boldsymbol{\sigma}_T(h')) = \boldsymbol{\pi}^2_T(\boldsymbol{\sigma}_T(h'))$. Then, for each node $h' \in H_T \cup Z$ with $h_1 \preceq h'$, $\boldsymbol{\pi}^3_T(\boldsymbol{\sigma}_T(h')) = 1$ if and only if $\boldsymbol{\pi}^2_T(\boldsymbol{\sigma}_T(h')) = 1$ because $\boldsymbol{\pi}^1_T(\boldsymbol{\sigma}_T(h_1)) = \boldsymbol{\pi}^2_T(\boldsymbol{\sigma}_T(h_1)) = 1$, and only one node is assigned the probability 1 in the corresponding information set $I_1$. It means that $\pi^3(I_1, a'_1) = \boldsymbol{\pi}^2_T(I_1, a'_1) = 1$ and $\pi^3(I_2, a_2) = \boldsymbol{\pi}^1_T(I_2, a_2) = 1$. Then, for each node $h' \in H_T$ with $\boldsymbol{\pi}^3_T(\boldsymbol{\sigma}_T(h')) = 1$, there is a node $B \in \mathcal{D}(\{\boldsymbol{\pi}^1_T, \boldsymbol{\pi}^2_T\})$ such that $h' \in B$ because $\mathcal{D}(\{\boldsymbol{\pi}^1_T, \boldsymbol{\pi}^2_T\})$ includes all nodes $h \in H_T$ with $\boldsymbol{\pi}^1_T(\boldsymbol{\sigma}_T(h)) = 1$ or $\boldsymbol{\pi}^2_T(\boldsymbol{\sigma}_T(h)) = 1$. That is, $\boldsymbol{\pi}^3_T \in \boldsymbol{\Pi}_T(\mathcal{D}(\{\boldsymbol{\pi}^1_T, \boldsymbol{\pi}^2_T\}))$. However, $\boldsymbol{\pi}^3_T \notin \{\boldsymbol{\pi}^1_T, \boldsymbol{\pi}^2_T\}$ and $\boldsymbol{\pi}^3_T \notin \Delta(\{\boldsymbol{\pi}^1_T, \boldsymbol{\pi}^2_T\})$ because any combination of $\boldsymbol{\pi}^1_T$ and $\boldsymbol{\pi}^2_T$ cannot represent $\boldsymbol{\pi}^3_T$ with $\boldsymbol{\pi}^3_T(I_1, a'_1) = 1$ and $\boldsymbol{\pi}^3_T(I_2, a_2) = 1$. $\square$

# D  MORE EXPERIMENTAL RESULTS

Tables 3 and 4 show the results on varying the target precision values on games $^3_3L^1_13$ and $^3_3L^1_14$. Solving a game with a smaller target precision value is similar to solving a larger game. In Tables 3 and 4, we can see that $\text{CG}_{\text{2random}}$ and $\text{CG}_{\text{2random}}^{\text{linrelax}}$ perform well in cases with larger target precision values but cannot perform well in cases with smaller target precision values because they add too many variables and constraints to the program for solving the restricted game at each iteration and then usually run out of memory. Other CG algorithms outperform our DCG algorithms in cases with larger target precision values, but they perform worse than our DCG algorithms in cases with smaller target precision values because they need too many iterations for convergence. Our DCG algorithms need significantly fewer indentations for the convergence in cases with smaller target precision values and then perform well. In addition, with more ranks (the game tree is wider, i.e., in $^3_3L^1_14$), our DCG with a pure BRO ($\text{DCG}_{\text{pure}}$ or $\text{DCG}_{\text{pure}}^{\text{linrelax}}$) performs better than the DCG with a randomized BRO ($\text{DCG}_{\text{random}}$ or $\text{DCG}_{\text{random}}^{\text{linrelax}}$). These results are consistent to our results shown in Section 4. Achieving

| | Runtime: $_3^3L_1^14$ with $\Delta U = 21$ | | | | | | | |
| Target Precision | $0.1\times\Delta U$ | $0.01\times\Delta U$ | $0.01\times\Delta U$ | $0.005\times\Delta U$ | $0.001\times\Delta U$ | 0.01 | $10^{-4}$ | $10^{-6}$ |
|---|---|---|---|---|---|---|---|---|
| $\text{DCG}_{\text{pure}}$ | 4.5s | 12s | **54s** | 88s | 177s | 206s | 358s | 381s |
| $\text{DCG}_{\text{pure}}^{\text{linrelax}}$ | 9s | 18s | 61s | **79s** | **104s** | **120s** | **162s** | **188s** |
| $\text{DCG}_{\text{random}}$ | 6.2s | **10s** | 61s | 100s | 249s | 302s | 501s | 537s |
| $\text{DCG}_{\text{random}}^{\text{linrelax}}$ | 11s | 25s | 66s | 90s | 117s | 134s | 191s | 191s |
| $\text{DCG}_{\text{2random}}$ | 17s | 29s | 139 | 230s | 506 | 991s | 75m | 76m |
| $\text{DCG}_{\text{2random}}^{\text{linrelax}}$ | 19s | 32s | 89s | 99s | 149s | 279s | 590s | 723s |
| $\text{CG}_{\text{pure}}$ | 4.5s | 13.3s | 270s | 18m | 93m | 138m | 372m | 434m |
| $\text{CG}_{\text{pure}}^{\text{linrelax}}$ | 8.3s | 29s | 340s | 712s | 38m | 51m | 129m | 151m |
| $\text{CG}_{\text{random}}$ | 3.4s | 13s | 216s | 764s | 150m | 206m | 557m | 10h |
| $\text{CG}_{\text{random}}^{\text{linrelax}}$ | 7.4s | 23s | 209s | 434s | 39m | 68m | 204m | 224m |
| $\text{CG}_{\text{2random}}$ | 1.8s | 72s | $\infty$ | $\infty$ | $\infty$ | $\infty$ | $\infty$ | $\infty$ |
| $\text{CG}_{\text{2random}}^{\text{linrelax}}$ | **1.6s** | 66s | $\infty$ | $\infty$ | $\infty$ | $\infty$ | $\infty$ | $\infty$ |
| | Iterations: $_3^3L_1^14$ with $\Delta U = 21$ | | | | | | | |
| Target Precision | $0.1\times\Delta U$ | $0.01\times\Delta U$ | $0.01\times\Delta U$ | $0.005\times\Delta U$ | $0.001\times\Delta U$ | 0.01 | $10^{-4}$ | $10^{-6}$ |
| $\text{DCG}_{\text{pure}}$ | 6 | 19 | 53 | 68 | 92 | 98 | 127 | 130 |
| $\text{DCG}_{\text{pure}}^{\text{linrelax}}$ | 3 | 8 | 20 | 23 | 28 | 31 | 38 | 42 |
| $\text{DCG}_{\text{random}}$ | 3 | 8 | 48 | 63 | 98 | 108 | 140 | 146 |
| $\text{DCG}_{\text{random}}^{\text{linrelax}}$ | 5 | 11 | 24 | 28 | 33 | 36 | 45 | 45 |
| $\text{DCG}_{\text{2random}}$ | 4 | 12 | 50 | 65 | 92 | 101 | 133 | 134 |
| $\text{DCG}_{\text{2random}}^{\text{linrelax}}$ | 2 | 5 | 14 | 15 | 19 | 23 | 29 | 31 |
| $\text{CG}_{\text{pure}}$ | 7 | 23 | 253 | 569 | 1562 | 2031 | 3394 | 4364 |
| $\text{CG}_{\text{pure}}^{\text{linrelax}}$ | 5 | 17 | 95 | 151 | 346 | 426 | 744 | 798 |
| $\text{CG}_{\text{random}}$ | 6 | 23 | 207 | 450 | 1476 | 1934 | 3935 | 4165 |
| $\text{CG}_{\text{random}}^{\text{linrelax}}$ | 6 | 13 | 68 | 110 | 308 | 392 | 756 | 816 |
| $\text{CG}_{\text{2random}}$ | 2 | 3 | - | - | - | - | - | - |
| $\text{CG}_{\text{2random}}^{\text{linrelax}}$ | 2 | 3 | - | - | - | - | - | - |

Table 4: Results on Leduc Poker $_3^3L_1^14$: $\infty$ means 'out of memory'.

a small target precision value is important because computing an accurate (or exact) solution is the core task of CG/double oracle algorithms (Bosansky et al., 2014).

## E  DETAILS ON LIMITATIONS

Runtime values reported in this paper for baselines may be different from the runtime values reported in previous papers (Zhang et al., 2021; Farina et al., 2021; Zhang et al., 2022c;b) because results reported in different papers may be obtained from different settings, e.g., different target precision values: $10^{-6}$ in this paper, $0.005 \times \Delta U$ (at least 0.03) in Zhang et al. (2022b), and $0.001 \times \Delta U$ (at least 0.006) in Zhang et al. (2022c); different program solvers: CPLEX in this paper and Gurobi (Farina et al., 2021; Zhang et al., 2022c;b); different implementations (the codes for these baselines are not available); and different computers, e.g., 2.3GHz CPU used in our paper but 2.80GHz CPU used in Farina et al. (2021), 16GB RAM used in our paper but 64GB or 60GB RAM used in Zhang et al. (2022c;b). If we directly compare our results with the result reported in previous papers for the same algorithm, we may obtain different conclusions: $\text{CG}_{\text{random}}^{\text{linrelax}}$ solves $_3^3L_1^13$ with target precision $10^{-6}$ by using 485s shown in Table 2, which is 203s reported in Farina et al. (2021). We may conclude that our implemented baseline $\text{CG}_{\text{random}}^{\text{linrelax}}$ is slower than the algorithm in Farina et al. (2021). However, $\text{CG}_{\text{random}}^{\text{linrelax}}$ solves $_3^3L_1^13$ with target precision $0.005 \times \Delta U$ by using 77s shown in Appendix D, which is 82s reported in Zhang et al. (2022b) for the algorithm in Farina et al. (2021). We may conclude that our implemented baseline $\text{CG}_{\text{random}}^{\text{linrelax}}$ is faster than the algorithm in Farina et al. (2021). Thus, to have a fair comparison with the previous baselines, all algorithms in our experiments are tested with the same setting. Overall, our results of baselines are consistent with the results reported in previous papers: normal-form CG algorithms perform well in games with shallow game trees but cannot perform well in games with deep game trees, and the baseline DAG performs well in games with narrow game trees but cannot perform well in games with wide game trees. Our DCG significantly overcomes their limitations.

