# OpenReview forum: "DAG-Based Column Generation for Adversarial Team Games"
_ICLR.cc/2024/Conference — Submitted to ICLR 2024_

### Official Review · Reviewer_toQL · 2023-10-12

**Soundness:** 3 good
**Presentation:** 3 good
**Contribution:** 3 good
**Rating:** 6
**Confidence:** 5

**Summary:**

The authors propose a column generation/double oracle-based framework for adversarial team games, in which the meta-Nash equilibrium is computed by allowing any (joint) action proposed by any of the strategies in the support through the TB-DAG framework, and the best responses are computed through an integer program. Via extensive experiments, the authors demonstrate that their method is more scalable than past techniques.

**Strengths:**

The techniques proposed by the paper are at least somewhat novel. The authors do a reasonable job of positioning the paper within prior work, addressing what aspects of their proposed techniques are new and performing ablation tests in the experiments. The experiments show that the method scales better than past methods, which is important for practical applications of TMECor.

**Weaknesses:**

As mentioned by the authors, the basic idea of using the sequence form to speed up column generation/double oracle for extensive-form games was proposed by Bosansky et al (2014). I do appreciate the authors' frank discussion of this comparison (Appendix A), but I think this should be reflected in the framing of the paper (e.g., perhaps Bosansky et al.'s method should be introduced in the body, and then the proposed DCG method should be framed as the natural analogue of that method for adversarial team games). I think the current framing of DCG makes it seem like it is more complex of a method than it really is. Phrased another way, in some sense I think Appendix A is the most important part of this paper from a conceptual standpoint, and it should be as much as possible moved into the body/discussed earlier, maybe even in the introduction. (if space is needed, for example, I'd suggest moving some of the ablation experiments, or the discussion of the BRO to the appendix; these matters are in my opinion less important than correct positioning of the work within past literature).

For the BRO, there should be some discussion of the total size of the representation (number of nonzeros in constraint matrix), as a function of e.g. the number of sequences, number of players, game size, etc. It seems to me that this size can be exponential in |H| in the worst case (when the number of players is large), which is somewhat troubling especially compared to past BROs which, as you say in Appendix A, do not have this problem. (For example, consider a polymatrix game with $N$ players and $A$ actions per player converted in the natural manner to extensive form---such a game would have $|H| = O((NA)^2)$ but every tuple of sequences will be relevant, so the  BRO will have representation size something like $A^N$).

On a more minor note, I would also like to see experiments on team-vs-team games, in which both teams would have to apply this column-generation method. In that setting, my guess would be the advantage over the baseline DAG algorithm will be less, because of the added inefficiency of having to perform CG for both sides---but I would still expect the method to perform very well.

**Questions:**

1. (from above) How does the proposed technique (or, the natural generalization thereof) perform in the setting of team-vs-team games?
1. Why does DCG_pure perform differently from DCG_linrelax^pure? Wouldn't an integer program solver first attempt to solve the linear relaxation anyway? (so they should do the same thing?)

---

> ### Author Response · Authors · 2023-11-18
> **To Reviewer toQL (1/2)**
>
> Thank you very much for your helpful comments. We have uploaded a new version of the paper to address your comments.
>
> **Comment**: About moving the related work in Appendix A to the main body of the paper.\
> **A**: Thanks for giving us this interesting idea. We once put all related work in the main body of the paper, but the space is not enough. So, we summarized the related work in Section 3 and presented details in Appendix A. Specifically, at the beginning of Section 3, we present:
>
> Solving Problem (1) is challenging because the team’s coordinated strategy space is exponential in the size of the game tree. With the procedure shown in Figure 1(c), CG can mitigate this challenge but still converges slowly in large games due to such a large strategy space. It has been shown that the TB-DAG-based algorithms are more efficient than CG in games with a small TB-DAG (Zhang et al., 2022b;c). However, solving Problem (1) through the TB-DAG is impractical in large games because the size of the TB-DAG is still exponential in the size of the game tree (Zhang et al., 2022c). To speed up, one straightforward idea is applying CG with the procedure shown in Figure 1(c) to the TB-DAG, which also is the direct application of the sequence-form double oracle (Bosansky et al., 2014) to the TB-DAG. That is, we compute a TMECor with a limited size of the TB-DAG, and then expand the DAG by computing a best response over the whole TB-DAG form strategy space, i.e., solving the problem: max_{\boldsymbol{x}} u_T (\boldsymbol{x}, \boldsymbol{y}_o). However, it is inefficient to compute a best response over the whole exponential-sized TB-DAG. As shown in experiments, this exponential size will make CG with the DAG-based BRO very inefficient. More details on related work are shown in Appendix A.
>
> **Comment**: About presenting the viewpoint of applying the sequence-form double oracle Bosansky et al. (2014) algorithm to ATGs.\
> **A**: Thanks again for your interesting idea. We once considered presenting the viewpoint of applying the sequence-form double oracle Bosansky et al. (2014) algorithm to ATGs. However, compared to the current viewpoint of applying CG to the TB-DAG, both viewpoints give us the same procedure and do not need the key features of the sequence-form double oracle algorithm in Bosansky et al. (2014), but using CG is the tradition in the mainstream literature on ATGs (Celli & Gatti, 2018; Farina et al., 2018; Zhang et al., 2021; Farina et al., 2021; Zhang et al., 2022b). Therefore, we chose the current viewpoint of applying CG to the TB-DAG:
> Recall that the sequence-form double oracle algorithm in Bosansky et al. (2014) is only for two-player extensive-form perfect-recall zero-sum games. However, as we mentioned, an ATG is an imperfect-recall game. Thus, the algorithm in Bosansky et al. (2014) cannot be directly applied to original ATGs. To apply the algorithm in Bosansky et al. (2014) for solving ATGs, we need to transform the original strategy representation of ATGs into the TB-DAG form. Then, we can compute a TMECor with a limited size of the TB-DAG for the restricted game and then expand the TB-DAG by computing a best response over the whole TB-DAG of the original game. Unfortunately, it is inefficient to compute a best response over the whole exponential-sized TB-DAG. Therefore, the sequence-form double oracle algorithm in Bosansky et al. (2014) that is directly applied to the TB-DAG is inefficient in large games. To improve the scalability, similar to Section 3, we can propose a novel sequence-form oracle algorithm framework that includes 1) computing a coordinated best response, 2) transforming this coordinated best response to the TB-DAG form, and then 3) adding this equivalent TB-DAG form strategy to the restricted game. This algorithm framework has an extra transformation step and needs a suitable BRO to reduce the cost of the transformation. We can see that the above algorithm procedure is the same as our current DCG. We chose to present our algorithm from the viewpoint of applying CG to the TB-DAG because:
>
> 1.	As we mentioned, both viewpoints will result in the same algorithm.
>
> 2.	In the mainstream literature on ATGs (Celli & Gatti, 2018; Farina et al., 2018; Zhang et al., 2021; Farina et al., 2021; Zhang et al., 2022b), CG was used to solve ATGs because the team’s strategy space is in normal form and is exponentially larger than the strategy space of the opponent (adversary).
>
> 3.	Because the strategy space of the opponent (adversary) is exponentially smaller than the team’s strategy space, we only expand the strategy space of the team. Thus, we do not need the key features of the sequence-form double oracle algorithm (Bosansky et al., 2014) in the restricted game (i.e., incomplete strategy and unallowed actions), as we mentioned in Appendix A.1.
>
> We added Appendix A.2 in the new version to discuss the viewpoint of applying the sequence-form double oracle Bosansky et al. (2014) algorithm to ATGs.

---

> > ### Author Response · Authors · 2023-11-18
> > **To Reviewer toQL (2/2)**
> >
> > **Comment**: About the representation size of BRO\
> > **A**: We have added Appendix A.3 in the new version to discuss the representation size of BRO. Specifically, for our pure BRO:
> > Our pure BRO ensures that only one joint action for each reachable belief will be used to expand the TB-DAG and then reduce the cost of the transformation. We exploit the team’s correlation property defined in Eq.(2a) (a probability flow) to improve the scalability of our BRO. In the original two-player von Stengel-Forges polytope (Von Stengel & Forges, 2008) for this probability flow, the constraints involve all relevant sequences. If we consider all relevant joint sequences for multiple team members, the number of constraints in our BRO will be exponential in the size of the game tree. To reduce the number of constraints, we consider only a subset of relevant joint sequences for nodes and sequences of the original game tree
> > (i.e., $\\boldsymbol{\Sigma}^\bowtie_T$) and limit the relevant relation of an information set and a joint sequence to $\\boldsymbol{\Sigma}^\bowtie_T$. That is, $\\boldsymbol{\Sigma}$$^\bowtie_T$= {$\sigma_T(h_i) \mid h_i\in H\cup Z\\}\cup\\{(\sigma_i,\times_{j\in T\setminus\{i\}}\varnothing)\mid\sigma_i\in\Sigma_i,i\in T\\}$ and   $I_i\bowtie\sigma_{T\setminus\{i\}}$   only if   $(\sigma_i(I_i),\sigma_{T\setminus\{i\}})\in \\boldsymbol{\Sigma}^\bowtie_T$.
> > We further add constraints in Eq.(2b) to ensure that $\xi(\sigma_T) =\prod_{i\in T}\xi(\sigma_T[i],\boldsymbol{\varnothing}_{-i})$ for   each $\sigma_T\in \\boldsymbol{\Sigma}^{\bowtie}_T$.
> > Therefore, our pure BRO shown in Program (3) has polynomial-sized constraints, i.e., $O(|H \cup Z||T |$) constraints.
> >
> > In the previous version, the constraints for our BRO were not clearly presented. Now, we have clearly presented these constraints in the new version and highlighted them in the paper.
> >
> > **Q**: (from above) How does the proposed technique (or, the natural generalization thereof) perform in the setting of team-vs-team games?\
> > **A**: First, in this paper, we focus on the ATGs with a team of players and an opponent, following the main mainstream literature on ATGs (Celli & Gatti, 2018; Farina et al., 2018; Zhang et al., 2021; Farina et al., 2021, Zhang et al. 2022a). Solving team-vs-team games could be the future work.
> >
> > Second, the baseline DAG directly solves the linear program for a TMECor after generating the whole TB-DAG. From results in Tables 1 and 2 for our current setting, we have seen that DAG is the fastest algorithm in games with very narrow game trees (the size of the TB-DAG is small) but runs out of memory in games with wide game trees (the size of the TB-DAG is large) where DAG performs worse than our algorithm. Similarly, we believe that in the setting of team-vs-team games, DAG will be faster than our algorithm in games with very narrow game trees but runs out of memory in games with wide game trees where DAG performs worse than our algorithm.
> >
> >
> >  **Q**: Why does DCG_pure perform differently from DCG_linrelax^pure? Wouldn't an integer program solver first attempt to solve the linear relaxation anyway? (so they should do the same thing?)\
> > **A**: As we mentioned, the setting of DCG^linrelax_pure is: at each iteration, it solves the linear relaxation of the BRO first to see if it can output the optimal solution added to the restricted game; if it cannot do that, it solves the original mixed-integer BRO and adds all feasible solutions for the BRO from the CPLEX solution pool to the original game.
> >
> > We use the default setting of the integer program solver, so we are not sure whether it first attempts to solve the linear relaxation. In addition to the linear relaxation, DCG^linrelax_pure usually adds more strategies to the restricted game than DCG_pure. As we mentioned in Section 4, the relaxed version DCG^linrelax_pure of DCG_pure, in most games with relatively narrow game trees, outperforms DCG_pure because it could reduce the cost of calling the mixed-integer BRO. However, in games with relatively wide game trees, DCG_pure outperforms DCG^linrelax_pure because DCG^linrelax_pure transforms too many coordinated strategies into the TB-DAG form, which incurs a very high cost in games with relatively wide game trees.

---

> > > ### Comment · Reviewer_toQL · 2023-11-18
> > >
> > > Thank you for the reply, especially for clarifying the BRO. My overall opinion of the paper remains positive, and I will keep my score.

---

> > > > ### Author Response · Authors · 2023-11-22
> > > >
> > > > Thank you very much for your positive response!

---

### Official Review · Reviewer_rs4T · 2023-10-31

**Soundness:** 4 excellent
**Presentation:** 3 good
**Contribution:** 3 good
**Rating:** 8
**Confidence:** 4

**Summary:**

This paper adapts the TB-DAG representation of an adversarial team game for column generation techniques, outperforming state-of-the-art CG implementations in numerical experiments on large instances.

**Strengths:**

I reviewed a previous version of this paper.

The paper's exposition has improved substantially from this previous iteration. Notation has been standardized to match what is used in existing literature, and the flow of the paper has been vastly improved. Additionally, a discussion has been added regarding the transformation cost of DCG based on using the BRO that has been proposed, which makes clear that the transformation cost is not overly expensive.

The paper provides a solid technical contribution in providing a novel CG framework for TMECor computation. Extensive experiments are provided to demonstrate the effectiveness of this framework and how it compares to SOTA approaches for TMECor computation. The authors sufficiently discuss the novelty of their approach in the main body of the paper as well as in Appendix A. It is clear that their approach outperforms existing approaches in certain settings, and thus is a valuable contribution.

**Weaknesses:**

Prior weaknesses that I (and other reviewers) had mentioned in an earlier review have been adequately addressed. In the past, some reviewers have mentioned concerns about the novelty of the approach; I think this has also been sufficiently addressed (see discussion in Strengths).

**Questions:**

N/A

---

> ### Author Response · Authors · 2023-11-18
> **To Reviewer rs4T**
>
> Thank you very much for your positive review.

---

### Official Review · Reviewer_cmYu · 2023-11-01

**Soundness:** 4 excellent
**Presentation:** 3 good
**Contribution:** 3 good
**Rating:** 6
**Confidence:** 4

**Summary:**

The paper proposes a novel algorithm for computing a TMEcor in adversarial team games. The main idea is that of combining the two algorithms currently used in the literature, Column Generation and DAG/belief-based representation. The proposed algorithm employs column generation for equilibrium computation on a reduced version of the game, while adopting the faster equilibrium computation enabled by the DAG. The role of the best response routine is to progressively expands the DAG representation.

The paper shows that this procedure does not worsen the theoretical bounds of the previous algorithms, and thoroughly evaluates different variants of the main algorithm on a large suite of games.

**Strengths:**

* clear positioning of the paper wrt the literature
* strong empirical results
* Comprehensive evaluation set and analysis
* I also appreciated the structure of the paper, where detailed and satisfactory discussions are moved to the appendix and the paper focuses on the main

**Weaknesses:**

* No clear winner between \emph{pure} or \emph{linrelax} versions of the algorithm
* the proopsed algorithm somewhat lack originality, in the sense that is an incremental improvement that recombines ideas already available in the literature. This lowers the contribution provided by the paper, even if my opinion is that this paper is interesting nonetheless.

**Questions:**

The paper was clear enough for me to understand the points
I suggest the following corrections/clarifications to improve the minor problems I found in specific points of the paper:
* in many occasions it is said that "by exploiting the team’s correlation property.. the BRO".. This property is not clear to me, and I thnk it should be better specified in the text
* the BRO algorithm presented in Section 3.3 is not completely novel. A similar algorithm was provided in "Subgame Solving in Adversarial Team Games" by Zhang et al., with the similar purpose of providing a more sparse BR (in that case sparsity is useful for reaching fewer private states in the same public state). My suggestion is to add a short reference to such a similar method.

---

> ### Author Response · Authors · 2023-11-18
> **To Reviewer cmYu**
>
> Thank you very much for your insightful comments. We have uploaded a new version of the paper to address your comments.
>
> **Comment**: No clear winner between \emph{pure} or \emph{linrelax} versions of the algorithm\
> **A**: As we mentioned, the setting of \emph{linrelax} is: at each iteration, it solves the linear relaxation of the BRO first to see if it can output the optimal solution added to the restricted game; if it cannot do that, it solves the original mixed-integer BRO and adds all feasible solutions for the BRO from the CPLEX solution pool to the original game.
>
> That is, \emph{linrelax} usually adds more strategies to the restricted game than \emph{pure}. Consider the relaxed version DCG^linrelax_pure of DCG_pure. As we mentioned in Section 4, the relaxed version DCG^linrelax_pure of DCG_pure, in most games with relatively narrow game trees, outperforms DCG_pure because it could reduce the cost of calling the mixed-integer BRO. However, in games with relatively wide game trees, DCG_pure outperforms DCG^linrelax_pure because DCG^linrelax_pure transforms too many coordinated strategies into the TB-DAG form, which incurs a very high cost in games with relatively wide game trees.
>
> **Comment**: About originality.\
> **A**: Our DCG does not just directly apply CG to the TB-DAG. Our work provides the first efficient TB-DAG CG algorithm, which includes a novel CG algorithm framework and solves many games that were previously unsolvable. Our DCG algorithm computes a pure coordinated best response (i.e., a coordinated strategy represented by a joint normal-form strategy of the team) in the original game tree first and then transforms it into the TB-DAG form. Our DCG has an extra transformation step, which does not exist in previous CG algorithms. Our DCG significantly overcomes the limitation of previous CG/double oracle algorithms for a TMECor with a novel CG algorithm framework and a suitable best response oracle. More details on related work are shown in Appendix A.
>
> We provide a solid technical contribution in providing a novel CG framework for TMECor computation. For example, we show that the transformation cost is only polynomial in the size of the original game tree, which makes it clear that the transformation cost is not overly expensive. We also show that our algorithm converges exponentially faster than the state-of-the-art CG algorithms.
>
> **Q**: in many occasions it is said that "by exploiting the team’s correlation property.. the BRO".. This property is not clear to me, and I thnk it should be better specified in the text\
> **A**: Thanks. We explicitly mentioned that the team’s correlation property is the probability flow defined in Eq.(2) in the new version.
>
> **Q**: the BRO algorithm presented in Section 3.3 is not completely novel. A similar algorithm was provided in "Subgame Solving in Adversarial Team Games" by Zhang et al., with the similar purpose of providing a more sparse BR (in that case sparsity is useful for reaching fewer private states in the same public state). My suggestion is to add a short reference to such a similar method.\
> **A**: Their technique is different from ours. In Appendix A.4 of the new version, we discussed the technique in "Subgame Solving in Adversarial Team Games" by Zhang et al.:
>
> Zhang et al. (2022a) provided a subgame technique to solve ATGs and proposed using CG for sparse solutions in the subgame solving algorithm. That is, they compute a best response starting from each reachable belief in each public state. To keep the number of these reachable beliefs small, they create sparse blueprints in the subgame solving algorithm by using a CG algorithm because “the support size of the blueprint generated by a CG algorithm, scales linearly with the number of iterations, which, under reasonable time constraints, rarely exceeds the hundreds” (Zhang et al., 2022a). Their sparse blueprint and our pure best response are both sparse solutions, but theirs is different from our suitable BRO:
>
>  1. The goals are different: Our goal is to reduce the cost of the transformation in our DCG, but their goal is to reduce the number of times that the BRO is called.
>
>  2. The sparsity concepts are different: Their sparse blueprint is about a small support size of the blueprint (a mixed strategy of the gadget game), but our pure best response is just about an action for each reachable state/belief.
>
>  3. The approaches are different: We propose a new suitable BRO, i.e., pure BRO, to improve our DCG, but they just directly use an existing CG algorithm for a sparse blueprint, i.e., they do not provide a new BRO algorithm.

---

> > ### Comment · Reviewer_cmYu · 2023-11-20
> >
> > Thanks for your detailed reviews. Your answers satisfy my questions, and I appreciated that you integrated the appendices with some extra details. I confirm my score.

---

> > > ### Author Response · Authors · 2023-11-22
> > >
> > > Thank you very much for your positive response!

---

### Official Review · Reviewer_yi8B · 2023-11-08

**Soundness:** 2 fair
**Presentation:** 2 fair
**Contribution:** 2 fair
**Rating:** 5
**Confidence:** 2

**Summary:**

This paper studies the problem of computing an equilibrium in team games in extensive form in which the team members can coordinate their strategies ex ante, before the beginning go the game. The paper introduces a new column generation approach working with the team-belief directed acyclic graph representation recently introduced by Zhang et al. (2022). The proposed method is experimentally evaluated and compared with state-of-the-art baselines in standard benchmark of games.

**Strengths:**

Providing better computational methods for computing equilibria in adversarial team games is of paramount importance in order to operationalize game-theoretic techniques in real-world settings beyond two-player zero-sum games.

The experimental evaluation provided in the paper is sufficiently broad.

**Weaknesses:**

I believe that the algorithm proposed in the paper is rather incremental over previous state-of-the-art techniques, for the following reasons:
1) The column generation algorithm doe not really add anything new compared to what done in classical column generation algorithms.
2) The only addition made by the proposed algorithm is related to the implementation of the best response oracle, and I believe this is not enough to constitute a contribution warrant publication in a top-tier AI conference as ICLR.

**Questions:**

None.

---

> ### Author Response · Authors · 2023-11-18
> **To Reviewer yi8B**
>
> Thank you very much for your valuable comments.
>
> **Comment**: About novelty and contribution:
>
> **A**: Our algorithm DCG does not just directly apply previous CG algorithms to the TB-DAG. Our work provides the first efficient TB-DAG CG algorithm, which includes a novel CG algorithm framework and solves many games that were previously unsolvable. Our DCG algorithm computes a pure coordinated best response (i.e., a coordinated strategy represented by a joint normal-form strategy of the team) in the original game tree first and then transforms it into the TB-DAG form. Our DCG has an extra transformation step, which does not exist in previous CG algorithms. Our DCG significantly overcomes the limitation of previous CG algorithms for a TMECor with a novel CG algorithm framework and a suitable best response oracle. More details on related work are shown in Appendix A.
>
> We provide a solid technical contribution in providing a novel CG framework for TMECor computation. For example, we show that the transformation cost is only polynomial in the size of the original game tree, which makes it clear that the transformation cost is not overly expensive. We also show that our algorithm converges exponentially faster than the state-of-the-art CG algorithms, and experimental results show that our algorithm is at least two orders of magnitude faster than the state-of-the-art baselines and solves games that were previously unsolvable.
>
> Thus, this paper creates a fundamental theory for applying the multiagent learning framework – Policy-Spaced Response Oracles (PSRO) (Lanctot et al., 2017) (a variant of CG) – to the TB-DAG for a TMECor.

---

### Meta-Review · Area_Chair_Qgv2 · 2023-12-06

**Metareview:**

This paper proposes a column-generation-based approach to solve for a coordinated strategy in an extensive form game. The paper develops a more computationally efficient method for integrating a team belief DAG with column generation. The reviewers identified the problem domain and experimental evaluations. Weaknesses included the novelty of the approach. One reviewer noted that the mathematical presentation of the methodology seems to be more complex than is necessary to convey the basic idea; as such the presentation may do more to obfuscate than to illuminate the reader as to the novelty of the contributions and their relationships with existing literature.

**Justification For Why Not Higher Score:**

The paper seems to be unnecessarily complex in its presentation of the mathematics. The text of table of results is condensed such that the text is nearly unreadable. A higher score would be justified by a clearer presentation.

**Justification For Why Not Lower Score:**

N/A

---

### Decision · Program_Chairs · 2024-01-16

Reject